# Promoters adopt distinct dynamic manifestations depending on transcription factor context

Anders S Hansen[1],* & Christoph Zechner[2,3,4],**

## Abstract

Cells respond to external signals and stresses by activating transcription factors (TF), which induce gene expression changes. Prior work suggests that signal-specific gene expression changes are partly achieved because *different* gene promoters exhibit distinct induction dynamics in response to the *same* TF input signal. Here, using high-throughput quantitative single-cell measurements and a novel statistical method, we systematically analyzed transcriptional responses to a large number of dynamic TF inputs. In particular, we quantified the scaling behavior among different transcriptional features extracted from the measured trajectories such as the gene activation delay or duration of promoter activity. Surprisingly, we found that even the *same* gene promoter can exhibit qualitatively distinct induction and scaling behaviors when exposed to different dynamic TF contexts. While it was previously known that promoters fall into distinct classes, here we show that the same promoter can switch between different classes depending on context. Thus, promoters can adopt context-dependent "manifestations". Our analysis suggests that the full complexity of signal processing by genetic circuits may be significantly underestimated when studied in only specific contexts.

**Keywords** Bayesian inference; manifestation; Msn2; promoter class switching; transcription factor dynamics
**Subject Categories** Chromatin, Transcription & Genomics; Computational Biology
**Mol Syst Biol.** (2021) 17: e9821

## Introduction

Exquisite regulation of gene expression underlies essentially all biological processes, including the remarkable ability of a single cell to develop into a fully formed organism. Transcription factors (TFs) control gene expression by binding to the promoters of genes and recruiting chromatin remodelers and the general transcriptional machinery. Recruitment of RNA Polymerase II enables the initiation of transcription, which produces mRNAs that are exported to the cytoplasm, where they are finally translated into proteins by the ribosome. Gene expression is primarily regulated at the level of promoter switching dynamics and initiation of transcription, which is associated with large cell-to-cell variability (Coulon *et al*, 2013). For practical reasons, however, gene expression is typically analyzed at the level of mRNAs (e.g., FISH) or proteins (e.g., immunofluorescence or GFP reporters) using bulk or single-cell approaches. Although powerful, these data provide only partial and indirect information about the underlying promoter states and transcription initiation dynamics. Moreover, although natural gene regulation is complex in both time (e.g., time-varying signals) and space (e.g., signaling gradients) (Li & Elowitz, 2019), experimental measurements tend to be limited to simple perturbations such as ON/OFF or dose–response curves under steady-state conditions.

Ideally, gene regulation should be studied at the level of promoter switching dynamics and transcription initiation events, using experimental approaches that capture gene expression in a sufficiently large number of single living cells in response to a broad range of dynamic inputs. Several studies have addressed some, but not all, of these challenges (Suter *et al*, 2011; Coulon *et al*, 2013; Hansen & O'Shea, 2013; Toettcher *et al*, 2013; Zoller *et al*, 2015). Here, through an integrated experimental and computational approach, we make a first attempt to realize this goal. We focus on a simple system, where a single inducible TF activates a target gene. Surprisingly, our approach reveals that even single gene promoters can display complex and counter-intuitive behaviors, which are difficult to explain by simple kinetic models. In particular, we show that genes exhibit "context-dependent manifestations", such that the same gene can switch between qualitatively different kinetic behaviors depending on which dynamic input it is exposed to. While it was previously known that promoters fall into distinct classes, we thus show here that the same promoter can switch class depending on context.

## Results

### Single-cell time-series measurements of promoter dynamics under complex TF inputs

To study how genes respond to complex and dynamic TF inputs, we focus on a large dataset that we previously generated (Fig EV1)

---

1   Department of Biological Engineering, Massachusetts Institute of Technology, Cambridge, MA, USA
2   Max Planck Institute of Molecular Cell Biology & Genetics, Dresden, Germany
3   Center for Systems Biology Dresden, Dresden, Germany
4   Cluster of Excellence Physics of Life, TU Dresden, Dresden, Germany
    *Corresponding author. Tel: +1 617 253 6086; E-mail: ashansen@mit.edu
    **Corresponding author. Tel: +49 351 210 2735; E-mail: zechner@mpi-cbg.de

(Hansen & O'Shea, 2013; Hansen & O'Shea, 2015) and which we have here converted from arbitrary fluorescence units to absolute protein abundances. In our setup, addition of a small molecule causes the budding yeast TF, Msn2, to rapidly translocate to the nucleus and activate gene expression (Fig 1A). Using microfluidics, rapid addition or removal of 1NM-PP1 allowed us to control both pulse length, pulse interval, and pulse amplitude of the TF (fraction of Msn2 that is activated) and simultaneously measure the single-cell response of natural and mutant Msn2 target genes using fluorescent reporters (Hansen & O'Shea, 2013; Hansen *et al*, 2015; Hansen & O'Shea, 2015) (Fig 1A).

We note that Msn2 naturally exhibits complex signal-dependent activation dynamics (Hao & O'Shea, 2012). First, Msn2 exhibits short pulses of nuclear localization in response to glucose starvation with dose-dependent frequency/number, and our pulse number/interval experiments were designed to match those (Fig 1B). Second, Msn2 largely exhibits a single pulse of nuclear localization in response to osmotic stress with dose-dependent duration, and our pulse duration experiments were designed to match this (Fig 1B). Third, Msn2 exhibits a sustained pulse of nuclear localization in response to oxidative stress with dose-dependent amplitude, and our amplitude-modulated experiments were designed to match this (Fig 1B) (Hao & O'Shea, 2012). In summary, we chose our TF inputs to be physiologically relevant. We note that the system is not subject to known feedback from Msn4 since Msn4 has been deleted in our system (Hao & O'Shea, 2012; Hansen & O'Shea, 2013; AkhavanAghdam *et al*, 2016), though we cannot rule out other forms of feedback. We also note that we replaced the target gene ORF with YFP and measured the endogenous gene response (Hansen & O'Shea, 2013; Hansen & O'Shea, 2015) and that the target genes are strictly Msn2-dependent (Hansen & O'Shea, 2013). Our extensive dataset contains 30 distinct dynamical Msn2 inputs for nine genes (270 conditions) and ∼ 500 cells per condition, numbering more than 100,000 single-cell trajectories in total (Fig 1B).

## Bayesian inference of promoter dynamics from time-lapse measurements

Gene promoters can generally exist in different transcriptionally active and inactive states (Coulon *et al*, 2013; Neuert *et al*, 2013). However, although our dataset is rich, since protein synthesis and degradation are slow processes, the raw YFP traces provide only indirect information about promoter state occupancies and dynamics. Bayesian methods provide an effective means to obtain statistical reconstructions of promoter states and transcription dynamics from time-lapse reporter measurements (Suter *et al*, 2011; Golightly & Wilkinson, 2011; Amrein & Künsch, 2012; Zechner *et al*, 2014). However, performing such reconstruction is computationally very demanding and existing approaches are typically too slow to handle large datasets like the one considered here, or rely on certain approximations which may be incompatible with the considered experimental system. To address this problem, we have developed a hybrid approach, which achieves accurate reconstructions while maintaining scalability.

Bayesian state reconstruction requires a mathematical model that captures the dynamics of the underlying molecular states and how those relate to the corresponding time-series measurements. To describe the dynamics of gene expression, we focus on a standard

Markov chain model, in which a promoter can switch between $L$ different states with distinct transcription rates $z_0, \ldots, z_{L-1}$ (e.g., transcriptionally inactive vs. active). Messenger RNA and protein YFP reporter copy numbers are described by two coupled birth-and-death processes. We account for extrinsic variability (Elowitz *et al*, 2002) at the translational level by considering the translation rate to be randomly distributed across a population of cells. The dynamic state of the overall gene expression system at time $t$ is denoted by $s(t) = (z(t), m(t), n(t))$, with $z(t) \in \{z_0, \ldots, z_{L-1}\}$ as the instantaneous transcription rate and $m(t)$ and $n(t)$ as the mRNA and YFP reporter copy numbers, respectively. We denote by $\mathbf{s}_{0:K} = \{s(t) | 0 \le t \le t_K\}$ a complete trajectory of $s(t)$ on the time interval $t \in [0, t_K]$. We consider a sequence of $K$ partial and noisy measurements $y_1, \ldots, y_K$ at times $t_1 < t_2 < \ldots < t_K$ along the trajectory. The statistical relationship between the measurements and the underlying state of the system is captured by a measurement density $p(y_k | s_k)$ with $s_k = s(t_k)$ for all $k = 1, \ldots, K$. In the scenario considered here, the measurements $y_1, \ldots, y_K$ represent noisy readouts of the reporter copy number extracted from time-lapse fluorescence movies. In order to infer $\mathbf{s}_{0:K}$ from a measured trajectory $y_1, \ldots, y_K$, we employ Bayes' rule, which can be stated as

$$p(\mathbf{s}_{0:K} | y_1, \ldots, y_K) \propto p(y_1, \ldots, y_K | \mathbf{s}_{0:K}) p(\mathbf{s}_{0:K}) = \prod_{k=1}^{K} p(y_k | s_k) p(\mathbf{s}_{0:K}),$$

(1)

with $p(\mathbf{s}_{0:K})$ as the prior probability distribution over trajectories $\mathbf{s}_{0:K}$, governed by the stochastic model of gene expression. The corresponding posterior distribution on the left-hand side captures the knowledge about a cell's trajectory $\mathbf{s}_{0:K}$ that we gain once we take into account the experimentally measured time series.

However, the posterior distribution in equation (1) is analytically intractable and one is typically left with numerical approaches. Sequential Monte Carlo (SMC) methods have been successfully applied to address this problem in the context of time-lapse reporter measurements (Zechner *et al*, 2014; Feigelman *et al*, 2016; Kuzmanovska *et al*, 2017). The core idea of these approaches is to generate a sufficiently large number of random sample paths $\mathbf{s}_{0:K}^{(i)}$ from the prior distribution and reweighing them using the measurement density $p(y_k | s_k)$ to be consistent with the posterior distribution. This is performed sequentially over individual measurement time points, which allows splitting the overall sampling problem into a sequence of smaller ones that can be solved more effectively (Methods and Protocols: Recursive Bayesian estimation).

The resulting SMC methods, however, are still computationally very expensive since the generation of an individual sample path $\mathbf{s}_{0:K}^{(i)}$ can span thousands or even millions of chemical events when considered on realistic experimental time scales. In the Msn2 induction system, for instance, trajectories often involve a large number of transcription and translation events, which would render conventional SMC approaches impractically inefficient. Alternatively, equation (1) can be calculated using analytical approximations (Huang *et al*, 2016). The main idea is to approximate the posterior distribution by a "simpler" distribution, such as a normal or log-normal distribution, which can be handled analytically. While analytical approximations can be substantially more efficient than SMC methods, the underlying distributional assumptions may not be suitable

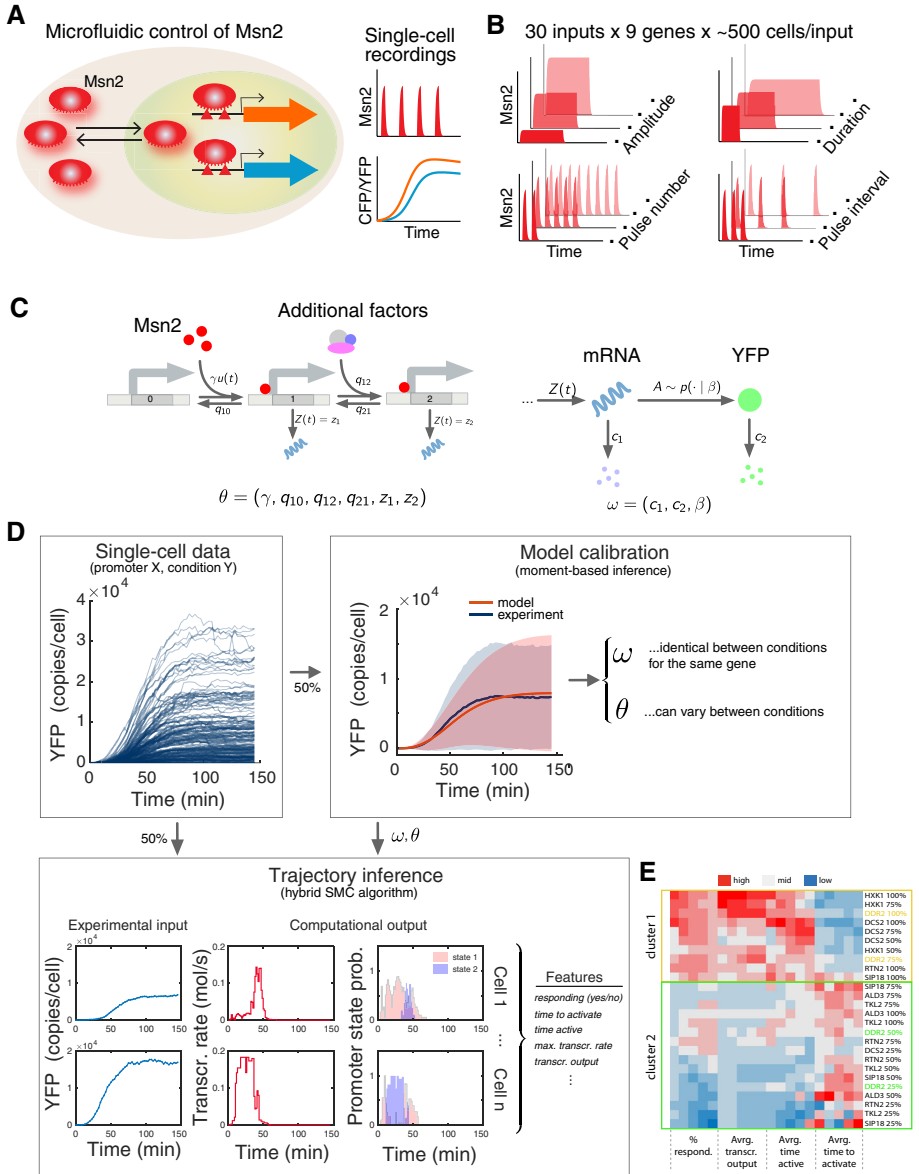

**Figure 1. Overview of Msn2 system and inference approach.**

A   Overview of microfluidic control of Msn2 activity and read-out of gene expression.

B   Overview of range of Msn2 input dynamics.

C   Stochastic model of gene expression. The promoter (left) can switch from its inactive state to its active state in an Msn2-dependent manner. Once active, mRNA can be transcribed at a certain rate $z_1$. Transcription can be further tuned by recruitment of additional factors, which is captured by a third state with distinct transcription rate $z_2$. Messenger RNA and protein dynamics are described as a two-stage birth-and-death process, accounting for extrinsic variability in the translation rate (right). A detailed description of the model can be found in Methods and Protocols: Stochastic model of Msn2-dependent gene expression).

D   Statistical reconstruction of promoter switching and transcription dynamics. Gene expression output trajectories were quantified for diverse Msn2 inputs in a large number of cells. One half of the trajectories was used to calibrate the model using a moment-based inference approach (Zechner *et al*, 2012). The model parameters corresponding to mRNA degradation, translation, and protein degradation where estimated once for each promoter from a single-pulse condition (50 min, 100% Msn2) but then held fixed for all other conditions. In contrast, the parameters corresponding to promoter switching and mRNA production where re-calibrated for each condition. The remaining half of the trajectories were used to reconstruct time-varying transcription rates and promoter state occupancies using the previously calibrated models in combination with the hybrid SMC algorithm (Methods and Protocols Hybrid sequential Monte Carlo). Several features characterizing the promoter and transcription dynamics were calculated from the single-cell reconstructions for all promoters and experimental conditions.

E   Hierarchical clustering of promoter dynamics. We considered all single-pulse experiments (10–50 min duration, 25–100% Msn2 induction, see (B) top row) for all promoters except the two *SIP18* mutants. For each condition, we calculated the percentage of responders, the average transcriptional output, the average time active, and the average time to activate. All features were averaged over five repeated runs of the inference pipeline. For a particular promoter and Msn2 induction level, we grouped together the respective features for all pulse lengths, giving rise to a 20-dimensional data point. In total, this leads to 28 20-dimensional data points (four Msn2 levels for seven promoters), which were normalized across individual features. Data points which had zero % responders for at least one of the pulse lengths were excluded from the analysis, since the remaining three features are not defined in this case. The data were clustered using a Euclidian distance metric and are shown as a heatmap, with cluster annotation.

in certain scenarios and lead to poor approximations. For instance, switch-like promoter dynamics are unlikely to be captured accurately by a continuous distribution such as a Gaussian. To address these problems, we developed a hybrid approach, which combines efficient analytical approximations with SMC sampling and thus strikes a balance between computational efficiency and accuracy. More precisely, only the promoter switching events have to be simulated stochastically, while the more expensive transcription and translation dynamics are eliminated from the simulation and handled analytically. This hybrid inference scheme targets the marginal posterior distribution

$$p(\mathbf{z}_{0:K}|y_1,\ldots,y_K) \propto p(y_1|\mathbf{z}_{0:1})\prod_{k=2}^{K}p(y_k|y_1,\ldots,y_{k-1},\mathbf{z}_{0:k})p(\mathbf{z}_{0:K}),\quad(2)$$

where the mRNA and reporter dynamics $m(t)$ and $n(t)$ have been integrated out. We derived expressions for the marginal likelihood functions $p(y_k|y_1,\ldots,y_{k-1},\mathbf{z}_{0:k})$ using an analytical approximation based on conditional moments (Methods and Protocols: Hybrid sequential Monte Carlo). The resulting method can be understood as a Rao-Blackwellized SMC approach (Doucet *et al*, 2000; Zechner *et al*, 2014). Using this hybrid approach, the sampling space can be significantly reduced, which makes inference efficient enough to deal with the large dataset considered in this study. A complete description of the method and a quantitative analysis of its accuracy based on simulated data can be found in Methods and Protocols and Fig EV2A and B.

## Inference of Msn2-dependent promoter and transcription dynamics

To quantify and understand how promoters respond to different dynamic TF inputs, we applied the hybrid SMC algorithm to the Msn2 datasets. To describe promoter activation and transcription, we focus on a canonical three-state promoter architecture (Fig 1C), which has been widely used in the literature (Coulon *et al*, 2013; Hansen & O'Shea, 2013). This model accounts for Msn2-dependent activation of the promoter after which mRNA can be transcribed at a certain rate. Transcription can be further tuned (for instance by recruitment of additional factors), which is captured by a third state with distinct transcription rate (Fig 1C).

The model involves a number of unknown parameters, which have to be determined prior to applying the hybrid SMC algorithm. To achieve this, we used a randomly selected subset of the Msn2 dataset in combination with an efficient moment-based approach, which reveals maximum *a posterior* estimates of the unknown parameters (Zechner *et al*, 2012). The inference was performed for each promoter and condition separately using 50% of the available single-cell trajectories. However, only the promoter switching and transcription rates were allowed to vary between conditions. The remaining parameters associated with mRNA degradation, translation, and protein degradation were estimated only for the first condition within experimentally constrained ranges (Hansen & O'Shea, 2013) and then held fixed for all other conditions (Methods and Protocols: Statistical inference of kinetic parameters).

The resulting calibrated models were then used to infer time-varying transcription rates and promoter state occupancies within individual cells from the remaining 50% of trajectories using the hybrid SMC algorithm (Fig 1D). From the large number of reconstructions, in turn, we computed a number of transcriptional features that summarize the single-cell expression dynamics of each promoter and condition (Methods and Protocols: Quantitative characterization of promoter dynamics). For the purpose of this study, we mainly focus on four transcriptional features. First, each cell was classified as responder or non-responder, depending on whether it was inferred to have resided in a promoter state with significant transcriptional activity for more than 2 min. For all responders, we estimated the time it took the promoter to switch into an active state (time to activate), the total time the promoter was in an active state (time active) as well as the integral over the time-varying transcription rate over the whole time course, which we refer to as transcriptional output. These dynamical features are chosen to be generic such that they do not rely on the particular structure of the considered promoter model. We remark that since the overall analysis pipeline depends on random number generation (e.g., splitting of data between model calibration and trajectory inference, Monte Carlo sampling), the inferred transcriptional features exhibit a certain degree of variability between repeated runs of the analysis. To quantify this variation, we performed five independent runs of the overall pipeline and calculated averages and standard errors. Data points shown in the following correspond to the inferred transcriptional features averaged across individual runs, unless stated otherwise. Both the calibrated models and temporal reconstructions were validated using a cross-validation approach (Fig EV2C–F). In summary, this combined experimental and computational approach allowed us to compare different promoters under a wide range of Msn2 contexts.

## Promoters exhibit context-dependent scaling behaviors and manifestations

To gain an overview of this high-dimensional dataset, we analyzed the gene expression responses to single pulses of nuclear Msn2 of different amplitudes (25, 50, 75, or 100%) for each promoter. Using hierarchical clustering, we uncovered the known promoter classes (Hansen & O'Shea, 2013) for most conditions (Fig 1E): slow activation, high amplitude threshold promoters (*SIP18*, *TKL2*) clustered together and fast activation, and low amplitude threshold promoters (*HXK1*, *DCS2*) also clustered together. Surprisingly, however, *DDR2* (Figs 1E and EV3) clustered with the slow, high threshold promoters at low Msn2 amplitudes (25, 50%), but with the fast, low threshold promoters at high Msn2 amplitudes (75, 100%). This suggests that the same promoter can switch promoter class and exhibit qualitatively different promoter and transcription dynamics when exposed to different Msn2 contexts.

To gain a better understanding of this phenomenon, we plotted the average time it took to activate the promoter (Fig 2A) and the average time the promoter was active (Fig 2B) against the transcriptional output for single-pulse inputs for *DDR2*. At low amplitude Msn2 input, the time it takes to activate *DDR2* for the first time increases with pulse length (Fig 2A), while both the time active (Fig 2B) and the transcriptional output increase only moderately (Fig 2A). In contrast, at high Msn2 amplitude, the time to activate appears fixed at approximately 5–10 min, but now transcriptional output and time active increase significantly with pulse duration.

This can be seen more clearly when plotting the dynamics of the inferred transcription rates of responding cells for low (Fig 2 C) and high (Fig 2D) Msn2 amplitudes. For 25% Msn2, the population-averaged transcription rate peaks at a time that scales with pulse length, while the maximum of the peak remains almost constant. This suggests that Msn2 duration predominantly regulates the probability to activate the promoter rather than the rate of transcription once the promoter becomes active. This behavior is in qualitative agreement with the slow activation, high amplitude threshold promoters such as *ALD3* (Fig 2E). In contrast, for 100% Msn2, the maximum transcription rate of *DDR2* increases by twofold to threefold between the 10 min and 50 min duration pulses, indicating that upon promoter activation, transcription can be further enhanced by the presence of Msn2. This behavior is characteristic for the fast activation, low amplitude promoters such as *DCS2* (Fig 2F).

In summary, this shows that a single promoter can switch between qualitatively distinct behaviors depending on Msn2 context. Here, *DDR2* behaves like one promoter class at low Msn2 amplitudes (pulse length regulates time to activate, but nothing else), but a distinct class at high Msn2 amplitudes (pulse length regulates time active, transcription output and maximum rate, but

not time to activate) (Fig 2G and H). While it is well known that promoters fall into distinct classes (Stavreva *et al*, 2009; Suter *et al*, 2011; Hao & O'Shea, 2012; Sharon *et al*, 2012; Hansen & O'Shea, 2013; Hansen & O'Shea, 2015; Haberle & Stark, 2018; King *et al*, 2020), what we show here is that the same promoter can switch from one class to another depending on context. To explain this phenomenon, we introduce the concept of "context-dependent manifestations". Operationally, we define a context-dependent manifestation of a promoter as a situation where the same promoter exhibits qualitatively distinct kinetic behaviors under different input contexts.

### Context-dependent promoter manifestations control gene expression noise

We next studied if promoters other than *DDR2* exhibit similar context-dependent promoter class switching. To this end, we analyzed the relationship between different promoter features under all input contexts and compared them with each other.

First, we analyzed the correlation between transcriptional output and the time the promoter was in any of the two transcriptionally permissive states (i.e., states 1 or 2 in Fig 1C) within

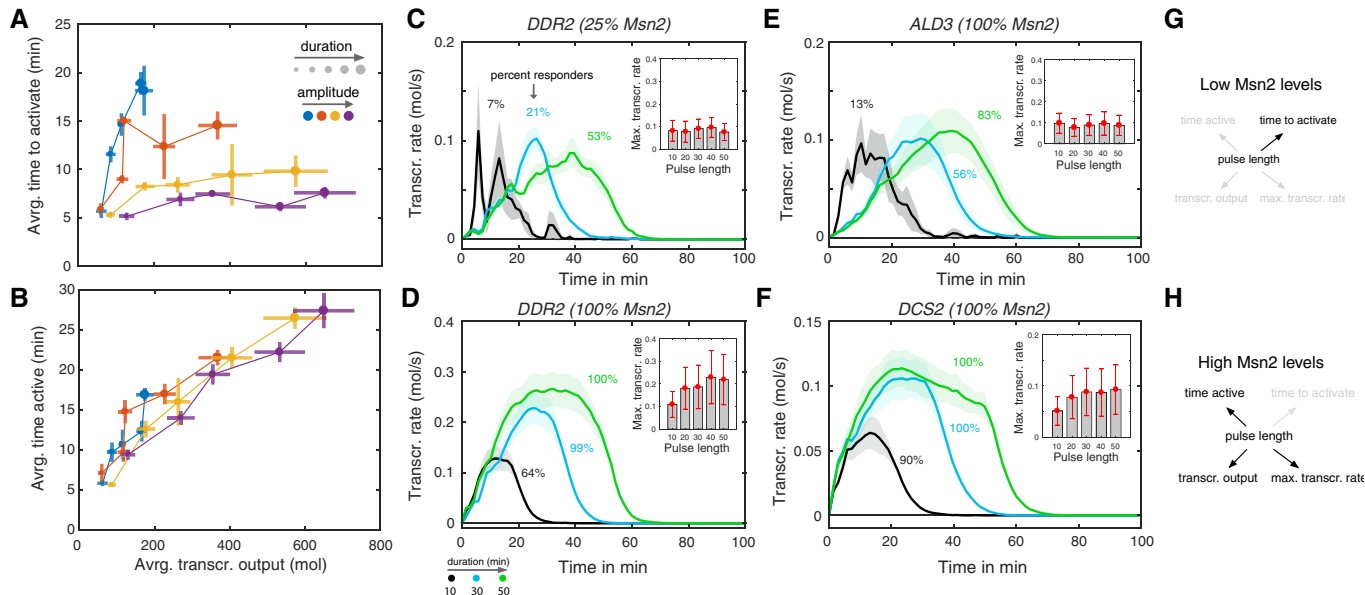

**Figure 2.  Context-dependent scaling behaviors.**

A, B  Scaling behaviors for *DDR2*. Scaling of time to activate (A) and total time active (B) for *DDR2* with transcriptional output. All three features were calculated as population averages across all responding cells per condition. Circles correspond to the mean of these features calculated over five independent inference runs and error bars indicate two times the standard error above and below the mean.

C, D  Population averages of the time-varying transcription rate were calculated for the 10, 30, and 50 min pulse conditions for 25% Msn2 amplitude (C) and 100% Msn2 amplitude (D) for *DDR2* considering only responding cells. Solid lines correspond to the mean calculated over five independent inference runs and shaded areas mark two times the standard error above and below the mean. The colored numbers indicate the estimated fraction of responding cells for the respective condition, averaged over all five inference runs. Inset plots show the maximum of the population-averaged transcription rate calculated over the whole time course. Circles correspond to means calculated over five independent inference runs and error bars mark two times the standard error above and below the means.

E, F  Time-varying transcription rates were calculated as in (C, D) for the 10, 30, and 50 min pulse conditions, and 100% Msn2 amplitude are shown for *ALD3* and *DCS2* for comparison.

G, H  Schematic model of *DDR2* promoter manifestations for low and high Msn2 induction levels. At Low Msn2 levels, Msn2 pulse length regulates the time to activate but not the other features. At High Msn2 levels, Msn2 pulse length regulates the time the promoter is active, transcriptional output, and maximal transcription rate, but it no longer regulates the time to activate.

individual cells (Fig 3A) for *TKL2*, *DDR2,* and *DCS2*. We refer to the latter as time transcribing. For *DCS2*, transcriptional output at the single-cell level shows a linear and nearly deterministic dependence on time transcribing. To validate this, we performed a regression analysis and found that a simple linear model where transcriptional output is proportional to time transcribing (with slope $k$) can explain most of the variation in transcriptional output ($R^2 \approx 1$; Fig 3A). Thus, for a given Msn2 amplitude, the effective rate of *DCS2* transcription is fixed and the single-cell transcriptional output can be determined very accurately by the time the promoter is in the transcriptionally permissive states. However, the rate of transcription is set by the Msn2 amplitude (i.e., $k$ increases with Msn2 amplitude). Thus, *DCS2* is remarkably

simple within the considered contexts and regulation by time transcribing and transcription rate can be decoupled. Similarly, for *DDR2*, the rate of transcription is also set by Msn2 amplitude. However, in comparison with *DCS2*, it exhibits larger variation for low and intermediate Msn2 amplitudes, which decrease toward higher Msn2 amplitudes. The inverse scaling of variability with amplitude can be explained by simple Markovian models with Msn2-dependent switching rates (Peccoud & Ycart, 1995; Hansen & O'Shea, 2013).

In contrast, *TKL2* resembles *DCS2* and *DDR2* at low Msn2 amplitudes ($R^2 \approx 1$), but at intermediate Msn2 amplitudes (Fig 3A, yellow), *TKL2* exhibits large variation, which decreases again for higher Msn2 amplitudes. Thus, surprisingly, time transcribing is a

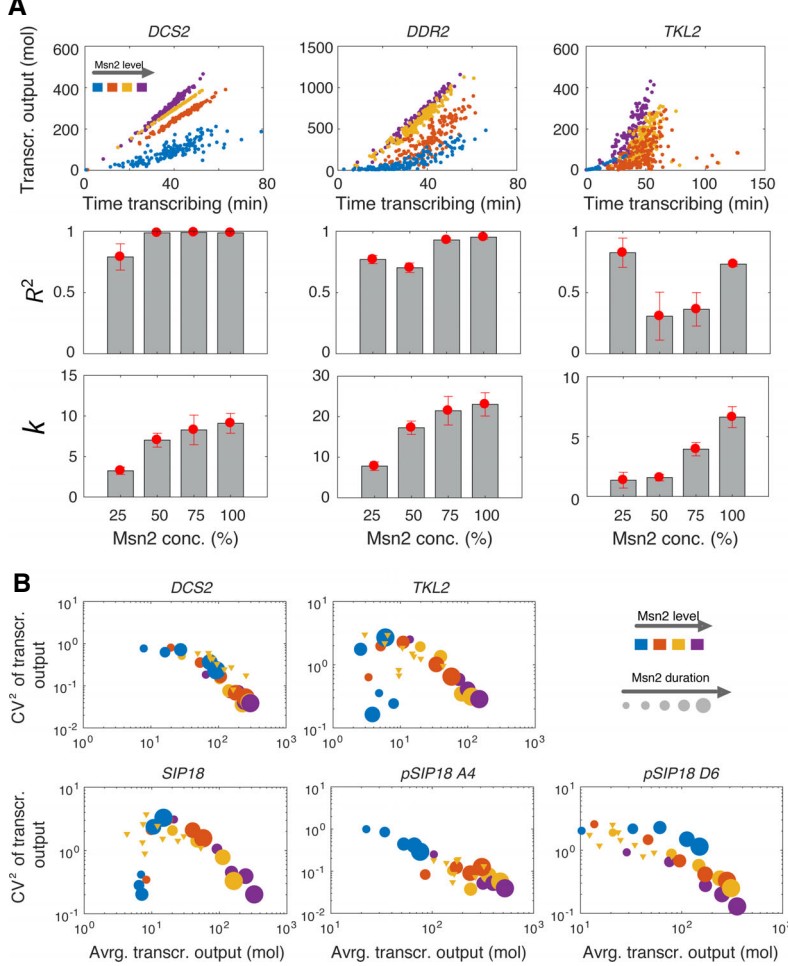

**Figure 3. Single-cell manifestations control gene expression noise.**

A Dependency of transcriptional output with time transcribing, defined as the time the promoter spends in any of the two transcriptionally permissive states (see Fig 1C). The left panel plots transcriptional output against time transcribing for individual cells for a 50 min pulse with 25, 50, 75, and 100% Msn2 input for all single-cell responses (responders and non-responders). Results are shown for one of the five independent inference runs. Linear regression analysis was performed to determine the $R^2$ and slope $k$ between transcr. output and time transcribing as shown in the center and bottom panels. Circles correspond to averages across five independent inference runs and error bars mark two times the standard error below and above the average.

B Scaling of noise with average transcriptional output for all Msn2 contexts. Noise is defined as the squared coefficient of variation of the transcriptional output calculated across individual cells. Single-pulse experiments of different Msn2 induction level and duration are shown as circles of varying size and color whereas all repeated-pulse experiments are shown as orange and equally sized triangles for visual clarity. Individual data points correspond to averages over five independent inference runs.

fairly poor predictor of *TKL2* transcriptional output at intermediate levels of Msn2 but a much better predictor at low and high Msn2 amplitudes. This non-monotonic relationship indicates that above a certain Msn2 concentration, additional promoter states with larger transcriptional activity become accessible, which increase in occupancy toward higher Msn2 amplitudes. This again suggests that the behavior of a single promoter can be dominated by distinct promoter architectures depending on input context.

The analysis above was concerned with the statistical relationship between time transcribing and transcriptional output in single cells for a single 50 min Msn2 pulse at different amplitudes. To generalize our analysis, we next studied how noise in transcriptional output (quantified using $CV^2 = \text{std}^2/\text{mean}^2$) scales with mean transcriptional output under all conditions (Fig 3B). As expected from previous studies (Bar-Even *et al*, 2006, Newman *et al*, 2006, Taniguchi *et al*, 2010), transcriptional noise uniformly decreases as transcriptional output increases for some genes such as *DCS2*. In contrast, *TKL2* and also *SIP18* exhibit more complex and non-monotonic noise scaling: low noise during low transcription, high noise during intermediate levels of transcription and again lower noise during high levels of transcription (Fig 3B), similar to the previous example in Fig 3A.

To further investigate this "inverse-U" scaling, we compared the behavior of the wild-type *SIP18* promoter with the two mutants A4 and D6 (Hansen & O'Shea, 2015) (Fig 3B). Mutant A4 resembles the simple inverse scaling relationship of *DCS2*. Similarly, mutant D6

also more closely resembles *DCS2*, albeit with a slightly weaker relationship between stronger expression and lower noise, suggesting that attenuation of this relationship can similarly be encoded in the promoter sequence. Taken together, these results demonstrate that modifying the number and location of Msn2 DNA binding sites in the promoter is sufficient to switch scaling and manifestation behavior.

### Memory-dependent promoter manifestations revealed by pulsatile Msn2 activation

We next analyzed how promoters respond to pulsatile Msn2 activation. Cells were exposed to four 5-min Msn2 pulses separated by 5, 7.5, 10, 15, or 20 min intervals. Some promoters behaved relatively simply, e.g., *DCS2* (Fig 4A). Most cells activate the *DCS2* promoter during the first pulse, and the promoter displays limited *positive memory* between pulses (Fig 4A). By *positive memory*, we refer to the fact that successive pulses of Msn2 activation increase the susceptibility of the promoter to become activated and induce higher gene expression. This has also been termed the *head-start* effect (Hao & O'Shea, 2012).

In contrast, the *SIP18* mutant D6 promoter (Hansen & O'Shea, 2015) exhibited very curious behavior: at 5-min intervals, there was significant positive memory (Fig 4A, top row). In contrast, with 20 min intervals, we observed *negative memory*: there was much lower expression during pulse 2–4, than during pulse 1(Fig 4A,

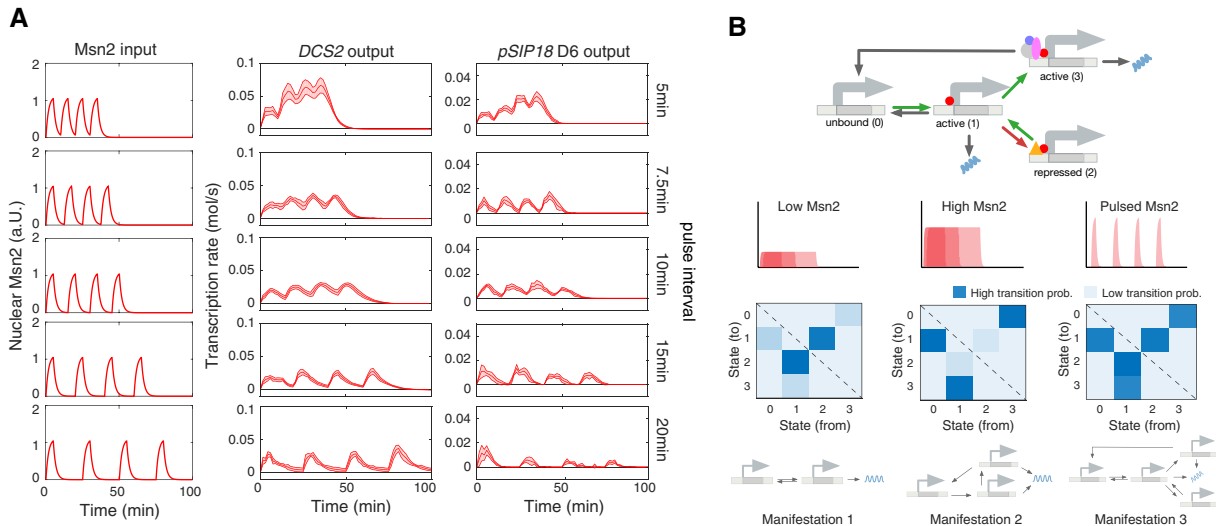

**Figure 4. Context-dependent promoter memory and model.**

A   Interval-dependent regulation of promoter memory. Cells where treated with four consecutive Msn2 pulses (75% induction level) with 5-min duration. The intervals between the pulses were 5, 7.5, 10, 15, and 20 min, respectively (left column). Population averages of the time-varying transcription rates were calculated for *DCS2* (middle column) and *SIP18* mutant D6 (right column) considering all cells per condition (responding and non-responding cells). Solid lines correspond to the mean of the population-averaged transcription rate calculated over five independent inference runs and shaded areas mark two times the standard error above and below the mean.

B   Toy model of context-dependent promoter manifestations. We considered a four-state promoter model with complex, nonlinear Msn2-dependent transition rates (top row). Green and red arrows indicate transitions, which are promoted or repressed by Msn2, respectively. Gray arrows correspond to Msn2-independent transitions. We simulated the promoter response to all thirty Msn2 inputs and quantified its dynamics by calculating the expected total number of transitions between all states (middle row heatmaps; blue show transitions with high probability (e.g., state 2 is rarely occupied in the middle scenario (High Msn2))). Depending on the Msn2 inputs, certain state transitions are favored, while others are effectively repressed. Therefore, different classes of dynamical inputs can reveal distinct manifestations of the same promoter (bottom row).

bottom row). In other words, exposure to one pulse of Msn2 inhibited transcription during subsequent pulses. Furthermore, comparing the different pulse intervals we observed a transition from positive memory at 5 and 7.5 min intervals to negative memory at 15 and 20 min intervals (Fig 4A).

While positive memory has previously been reported (Hao & O'Shea, 2012; Hansen & O'Shea, 2013), a context-dependent switch from positive to negative memory has not. We note that a sharp transition from positive to negative promoter memory is difficult to explain by simple kinetic models and that this type of behavior only becomes visible once the response to diverse dynamic inputs are analyzed. Although the underlying molecular mechanism is unknown, we show in Fig EV4 a hypothetical toy model that could explain such a switch from positive to negative memory. In conclusion, these data provide another example of how the same promoter can exhibit very different quantitative and qualitative behaviors depending on the context—in this case, depending on the interval between Msn2 pulses.

## Discussion

Here, we quantitatively analyze the dynamic input–output relationship in a simple inducible gene regulation system. Previously, a large number of studies have shown that promoters fall into distinct classes (e.g., fast vs. slow; low vs. high threshold) and that different promoters decode dynamic stimuli differently (Stavreva et al, 2009; Suter et al, 2011; Hao & O'Shea, 2012; Sharon et al, 2012; Hansen & O'Shea, 2013; Hansen & O'Shea, 2015; Haberle & Stark, 2018; King et al, 2020). For example, a slow promoter may filter out a brief and transient stimulus (Purvis & Lahav, 2013). However, promoter class was assumed to be a fixed property.

Here, we show that promoters can switch between distinct classes depending on context. We show that even under these relatively simple conditions, the same promoter can exhibit context-dependent scaling and induction behaviors (Figs 1–4 and EV5). To describe this observation, we introduce the concept of context-dependent manifestations. The underlying number of molecular states of a promoter is potentially enormous; if we were to enumerate the combinatorial number of states based on nucleosome positions, TF occupancy at each binding site, binding of co-factors such as Mediator, SAGA, TFIID, RNA Polymerase II, and numerous other factors, the number of discrete molecular states would be astronomically high. When we measure a dose–response, we may observe only certain rate-limiting regimes or manifestations of the system. What we show here is that the particular observed manifestation can be highly context-dependent and very distinct quantitative behaviors can be observed under different contexts even in systems that are seemingly simple.

Does this mean that the concept of a few discrete promoter states is too strong an approximation to be useful? We suggest that this is not necessarily the case. Our analyses show that for a given context, a 3-state promoter architecture was capable of quantitatively describing promoter dynamics. However, the specific three promoter states and their associated rates were in general dependent on Msn2 context. In other words, a complicated system can manifest itself in a simpler form under specific conditions. Comparing

different manifestations across multiple input contexts can thus help to unravel the overall complexity of promoter dynamics.

To illustrate this point further, consider a hypothetical promoter with four major states (Fig 4B). If under some dynamical Msn2 inputs, this promoter reduces to simpler architectures (e.g., 2-state), but not under other Msn2 inputs (e.g., remains 4-state), then the observed quantitative manifestation of the promoter is dependent on Msn2-context. To more concretely demonstrate an example of this, we performed simulations of a complex 4-state promoter with nonlinear Msn2-dependent switching rates (Fig 4B; see Methods and Protocols: Toy model of a complex, context-dependent promoter for details on the model) to all thirty dynamical Msn2 inputs. To characterize the dynamics of the promoter, we calculated the average number of transitions between all promoter states. These results show that depending on the Msn2 inputs, certain state transitions are favored, while others are effectively repressed. In particular, the same promoter can behave effectively like a 2-, 3-, or 4-state promoter, depending on which type of dynamical input it is exposed to (Fig 4B). Mechanistically, one could imagine a promoter state that requires sustained chromatin remodeling and only becomes available if the Msn2 pulse and concentration is sufficiently high, as we previously suggested for *SIP18* and its two promoter mutants studied here (Hansen & O'Shea, 2015). But this is speculative, and the precise molecular mechanisms underlying the distinct promoter manifestations observed here remain unknown. We suggest elucidating the molecular mechanisms underlying promoter manifestations as an important area for future research.

Our results have two important potential implications. First, our results suggest that system identification efforts based on limited sets of experimental conditions within complex systems are unlikely to be successful in the sense of capturing the full range of relevant behaviors of the underlying molecular pathways. In extreme cases, we may arrive at different and possibly contradictory conclusions about a pathway's inner workings depending on which experimental context we choose to study. The only solution to this problem is to resort to experimental and computational approaches that capture a pathway's response to a sufficiently broad range of physiologically meaningful contexts. Much more work on simple systems will be necessary to truly understand the relevant complexity of signal processing in cells, and we hope the approaches developed here will be helpful in this regard.

Second, a major conundrum in quantitative biology has been how to reconcile the remarkable spatiotemporal precision of biological systems with the high degree of gene expression noise observed at the single-cell level (Elowitz et al, 2002; Cai et al, 2006; Li & Elowitz, 2019). For example, when information transduction capacities have been measured for simple pathways, such systems appear to be barely capable of reliable distinguishing ON from OFF ($\sim 1$ bit) (Cheong et al, 2011; Uda et al, 2013; Selimkhanov et al, 2014; Voliotis et al, 2014). Since these studies were done under strict experimental conditions, they may have captured only one out of multiple manifestations. Our results suggest that if all physiologically relevant manifestations could be captured, the estimated information transduction capacity of biochemical pathways could be substantially greater than previously estimated. This could, in part, explain the remarkable signal processing capabilities of biological systems.

# Materials and Methods

## Reagents and Tools Table

| Reagent/Resource | Reference or source | Identifier or catalog number |
|---|---|---|
| **Experimental models** | | |
| *TPK1*$^{M164G}$ *TPK2*$^{M147G}$ *TPK3*$^{M165G}$ *msn4Δ*::TRP1/LEU2 *MSN2*-mCherry *NHP6a*-iRFP::kanMX *hxk1*::mCitrineV163A/SCFP3A-spHIS5 (Diploid) | Hansen and O'Shea (2013) | EY2810 |
| *TPK1*$^{M164G}$ *TPK2*$^{M147G}$ *TPK3*$^{M165G}$ *msn4Δ*::TRP1/LEU2 *MSN2*-mCherry *NHP6a*-iRFP::kanMX *sip18*::mCitrineV163A/SCFP3A-spHIS5 (Diploid) | Hansen and O'Shea (2013) | EY2813 |
| *TPK1*$^{M164G}$ *TPK2*$^{M147G}$ *TPK3*$^{M165G}$ *msn4Δ*::TRP1/LEU2 *MSN2*-mCherry *NHP6a*-iRFP::kanMX *rtn2*::mCitrineV163A/SCFP3A-spHIS5 (Diploid) | Hansen and O'Shea (2013) | EY2816 |
| *TPK1*$^{M164G}$ *TPK2*$^{M147G}$ *TPK3*$^{M165G}$ *msn4Δ*::TRP1/LEU2 *MSN2*-mCherry *NHP6a*-iRFP::kanMX *dcs2*::mCitrineV163A/SCFP3A-spHIS5 (Diploid) | Hansen and O'Shea (2013) | EY2819 |
| *TPK1*$^{M164G}$ *TPK2*$^{M147G}$ *TPK3*$^{M165G}$ *msn4Δ*::TRP1/LEU2 *MSN2*-mCherry *NHP6a*-iRFP::kanMX *tkl2*::mCitrineV163A/SCFP3A-spHIS5 (Diploid) | Hansen and O'Shea (2013) | EY2822 |
| *TPK1*$^{M164G}$ *TPK2*$^{M147G}$ *TPK3*$^{M165G}$ *msn4Δ*::TRP1/LEU2 *MSN2*-mCherry *NHP6a*-iRFP::kanMX *ddr2*::mCitrineV163A/SCFP3A-spHIS5 (Diploid) | Hansen and O'Shea (2013) | EY2825 |
| *TPK1*$^{M164G}$ *TPK2*$^{M147G}$ *TPK3*$^{M165G}$ *msn4Δ*::TRP1/LEU2 *MSN2*-mCherry *NHP6a*-iRFP::kanMX *ald3*::mCitrineV163A/SCFP3A-spHIS5 (Diploid) | Hansen and O'Shea (2013) | EY2828 |
| *TPK1*$^{M164G}$ *TPK2*$^{M147G}$ *TPK3*$^{M165G}$ *msn4Δ*::TRP1/LEU2 *MSN2*-mCherry *NHP6a*-iRFP::*KAN sip18*::mCitrine_V163A/SCFP3A-spHIS5 *pSIP18* Mutant A4 with 4 STREs (Diploid) | Hansen and O'Shea (2015) | EY2967 |
| *TPK1*$^{M164G}$ *TPK2*$^{M147G}$ *TPK3*$^{M165G}$ *msn4Δ*::TRP1/LEU2 *MSN2*-mCherry *NHP6a*-iRFP::*KAN sip18*::mCitrine_V163A/SCFP3A-spHIS5 *pSIP18* Mutant D6 with 6 STREs (Diploid) | Hansen and O'Shea (2015) | EY2996 |
| **Chemicals, enzymes, and other reagents** | | |
| 1-NM-PP1 | Hansen and O'Shea (2013) | 1-NM-PP1 |
| **Software** | | |
| Image analysis code | Hansen *et al* (2015) | https://www.nature.com/articles/nprot.2015.079 |
| Code and raw data to reproduce all plots in this manuscript | This study | https://github.com/zechnerlab/PromoterManifest/ |
| **Other** | | |
| Gene expression data for *ALD3, TKL2, DCS2, DDR2, HXK1, RTNA*, and *SIP18* | Hansen and O'Shea (2013) | https://www.embopress.org/doi/10.1038/msb.2013.56 |
| Gene expression data for *pSIP18* mutant A4 and D6 | Hansen and O'Shea (2015) | https://www.sciencedirect.com/science/article/pii/S2211124715007950 |
| Compilation of all single-cell trajectories used in this study | This study | https://zenodo.org/record/2755026 |

## Methods and Protocols

### Overview of experiments and source data

We note that the data used here were acquired previously (Hansen & O'Shea, 2013; Hansen & O'Shea, 2015), but in the interest of making it clear how the experiments were conducted, we provide a brief outline of the experimental setup in the sections below. The data in concentration units of arbitrary fluorescence were previously acquired and described (Hansen & O'Shea, 2013; Hansen & O'Shea, 2015). Here, we used absolute abundance quantification (Huang *et al*, 2016) to convert the data to absolute numbers of YFP and CFP proteins per cell. All the source data supporting this manuscript are freely available together with a detailed ReadMe file at https://zenodo.org/record/2755026. Information about the yeast strains can be found in the Reagent and Tools Table.

### Microfluidics and time-lapse microscope

Since the unnormalized data were previously acquired, here we only briefly describe the experimental methods. Microfluidic devices were constructed as previously described (Hansen & O'Shea, 2013). We furthermore refer the reader to a detailed protocol describing how to construct microfluidic devices and computer code for controlling the solenoid valves (Hansen *et al*, 2015). Briefly, for microscopy experiments, diploid yeast cells were grown overnight at 30°C with shaking at 180 RPM to an OD600 nm of ca. 0.1 in low fluorescence medium without leucine and tryptophan, quickly collected by suction filtration and loaded into the five channels of a microfluidic device pretreated with concanavalin A (4 mg/ml). The setup was mounted on an inverted fluorescence microscope kept at 30°C. The microscope automatically maintains focus and acquires phase-contrast, YFP,

CFP, RFP, and iRFP images from each of five microfluidic channels for 64 frames with a 2.5 min time resolution corresponding to imaging from −5 to 152.5 min. Solenoid valves control delivery of 1-NM-PP1 to each microfluidic channel. For full details on the range of input conditions, please see source data at https://zenodo.org/record/2755026.

### Image analysis and YFP quantification and normalization

Time-lapse movies were analyzed using custom-written software (MATLAB) that automatically segments yeast cells based on phase-contrast images and tracks cells between frames. The image analysis software and a protocol describing how to use it is available elsewhere (Hansen *et al*, 2015). The arbitrary fluorescence units were converted to absolute abundances by comparing fluorescence to strains with known absolute abundances and by segmenting the cell to calculate the total number of YFP molecules per cell per timepoint (Huang *et al*, 2016). Maturation delay was accounted for by shifting the YFP trajectories by a fixed time interval of 12.5 min, corresponding to the first five measurement time points.

### Quantification of nuclear Msn2 dynamics

Msn2 was visualized as an Msn2-mCherry fusion protein. This allows accurate quantification of the nuclear concentration of Msn2 over time (Msn2 only activates gene expression when nuclear) as previously described (Hao & O'Shea, 2012; Hansen & O'Shea, 2013). From the resulting time courses, we extracted continuous functions $u(t)$, which served as inputs to our stochastic promoter model. Since we found nuclear Msn2 concentration to vary very little between cells (Fig EV1), we considered $u(t)$ to be deterministic. We performed this as described previously (Hansen & O'Shea, 2013) and elaborated on here. We model nuclear Msn2 import with first-order kinetics:

$$u(t) = u_0(1 - e^{-k_1 t}). \qquad (3)$$

That is, if Msn2 is cytoplasmic at time $t = 0$, the nuclear level of Msn2 at a later time $t$ is given by the above expression where $u_0$ is the maximal level of nuclear Msn2 for the given concentration of 1-NM-PP1. We chose the 1-NM-PP1 concentrations as 100, 275, 690, and 3,000 nM such that they would correspond to approximately 25, 50, 75, and 100 of maximal nuclear Msn2. The parameter $k_1$ is a fit parameter describing the rate of nuclear import, which we found to vary slightly depending on the 1-NM-PP1 concentration. Similarly, we model export of Msn2 from the nucleus as a first-order process:

$$u(t_2) = u(t_1)e^{-k_2(t_2 - t_1)}. \qquad (4)$$

Here, $u(t_1)$ is the nuclear level of Msn2 when the microfluidic device was switched to medium without 1-NM-PP1. Correspondingly, $u(t_2)$ is the nuclear level of Msn2 at some later time $t_2 > t_1$. This is to account for the fact that, depending on the pulse duration, Msn2 may not have reached its maximal nuclear level, $u_0$. The parameters $u_0$, $k_1$ and $k_2$ were determined through fitting. Specifically, we took the full 30 different pulses and inferred the best-fit values for $u_0$, $k_1$, and $k_2$ using least squares fitting. The values are shown below:

| [1-NM-PP1] (nM) | $u_0$ | $k_1 (min^{-1})$ | $k_2 (min^{-1})$ |
|---|---|---|---|
| 100 | 313.2 | 1.11 | 0.97 |
| 275 | 774.5 | 0.61 | 0.81 |
| 690 | 1,107.8 | 0.59 | 0.57 |
| 3,000 | 1,410.1 | 1.07 | 0.29 |

### Stochastic model of Msn2-dependent gene expression

We describe Msn2-dependent gene expression using a canonical three-state model as shown in Fig 1C. The promoter is described as a continuous-time Markov chain, which switches stochastically between three states of different transcriptional activity. Correspondingly, the rate of transcription at time $t$ is governed by a stochastic process $Z(t) \in \{z_0, z_1, z_2\}$, whose value changes discontinuously whenever the promoter transitions from one state into another. In the absence of nuclear Msn2, the promoter is in its transcriptionally inactive state ($z_0 = 0$), where no transcripts are produced. Upon recruitment of Msn2 to the promoter, it can switch into a transcriptionally permissive state in which transcription takes place with propensity $z_1$. To account for Msn2-dependent promoter activation, we consider the switching rate from $z_0$ to $z_1$ to depend on the nuclear Msn2 abundance. For simplicity, we consider a linear dependency, i.e., $q_{01}(t) = \gamma u(t)$, with $u(t)$ as the Msn2 abundance at time $t$. The corresponding reverse rate $q_{10}$ is considered to be constant. We assume that transcription can be further enhanced by recruitment of additional factors such as chromatin remodeling complexes and general transcriptional factors. This is captured in our model by introducing a third state with transcription rate $z_2$ and corresponding transition rates $q_{12}$ and $q_{21}$. With this, we can describe the time-dependent probability distribution over the transcription rate $P_Z(t) = (P(Z(t) = 0|\theta), P(Z(t) = z_1|\theta), P(Z(t) = z_2|\theta))^T$ in terms of a forward equation.

$$\frac{d}{dt}P_Z(t) = Q(t)P_Z(t) = \begin{pmatrix} -q_{01}(t) & q_{10} & 0 \\ q_{01}(t) & -q_{10}-q_{12} & q_{21} \\ 0 & q_{12} & -q_{21} \end{pmatrix} P_Z(t), \quad (5)$$

with $P_Z(0) = p_{z,0}$ as some initial distribution over $Z(t)$ and $\theta = \{\gamma, q_{10}, q_{12}, q_{21}, z_1, z_2\}$ as a set of parameters. In the following, we denote by $\mathbf{z}_t = \{z(s)|0 \le s \le t\}$ a complete realization of $Z(t)$ on a fixed time interval $[0,t]$. Furthermore, we introduce the conditional path distribution $p(\mathbf{z}_t|\theta)$ which measures the likelihood of observing a particular trajectory $\mathbf{z}_t$ for a given parameter set $\theta$. Note that it is straightforward to draw random sample paths $\mathbf{z}_t$ from this distribution using Gillespie's stochastic simulation algorithm (SSA) (Gillespie, 2007) or its variants.

Transcription and translation are modeled as a two-stage reaction network as shown in Fig 1C. We denote by $M(t)$ and $N(t)$ the copy numbers of mRNA and protein at time $t$, respectively. The parameters $c_1$ and $c_2$ are the mRNA and protein degradation rates and $A$ is the protein translation rate. To account for cell-to-cell variability in protein translation, we consider the latter to be randomly distributed across isogenic cells, i.e., $A \sim p(a|\beta)$, with $p(a|\beta)$ as an arbitrary probability density function (pdf) with positive support and $\beta$ as a set of hyperparameters characterizing this distribution

(Zechner *et al*, 2012; Zechner *et al*, 2014). Here, we consider as hyperparameters the average and coefficient of variation (CV) of $A$ such that $\beta = \{\langle A \rangle, CV[A]\}$. Consequently, $\beta$ captures the magnitude and variability associated with protein translation. In the following, we denote by $\omega = \{c_1, c_2, \beta\}$ the set of parameters corresponding to transcription and translation.

For a given set of parameters $\theta$ and $\omega$ and a concrete realization of the translation rate $A$, the overall dynamics of the joint system state $(Z(t), M(t), N(t))$ can be described by a Markov chain. However, due to the random variability over $A$, each cell is associated with a differently parameterized Markov chain. This results in a heterogeneous Markov model, whose computational analysis turns out to be challenging (Zechner *et al*, 2014). One way to address this issue is to augment the state space by the random variable $A$ and to formulate a master equation on this extended space. For $S(t) = (Z(t), M(t), N(t), A)$, such master equation reads

$$
\begin{aligned}
\frac{d}{dt}P(z_0, m, n, a, t) &= z_0 P(z_0, m-1, n, a, t) + c_1(m+1)P(z_0, m+1, n, a, t)\\
&\quad + amP(z_0, m, n-1, a, t) + c_2(n+1)P(z_0, m, n+1, a, t)\\
&\quad - [z_0 + c_1 m + am + c_2 n]P(z_0, m, n, a, t)\\
&\quad + q_{10}P(z_1, m, n, a, t) - q_{01}(t)P(z_0, m, n, a, t)\\
\frac{d}{dt}P(z_1, m, n, a, t) &= z_1 P(z_1, m-1, n, a, t) + c_1(m+1)P(z_1, m+1, n, a, t)\\
&\quad + amP(z_1, m, n-1, a, t) + c_2(n+1)P(z_1, m, n+1, a, t)\\
&\quad - [z_1 + c_1 m + am + c_2 n]P(z_1, m, n, a, t)\\
&\quad + q_{01}(t)P(z_0, m, n, a, t) - q_{10}P(z_1, m, n, a, t)\\
&\quad + q_{21}P(z_2, m, n, a, t) - q_{12}P(z_1, m, n, a, t)\\
\frac{d}{dt}P(z_2, m, n, a, t) &= z_2 P(z_2, m-1, n, a, t) + c_1(m+1)P(z_2, m+1, n, a, t)\\
&\quad + amP(z_2, m, n-1, a, t) + c_2(n+1)P(z_2, m, n+1, a, t)\\
&\quad - [z_2 + c_1 m + am + c_2 n]P(z_2, m, n, a, t)\\
&\quad + q_{12}P(z_1, m, n, a, t) - q_{21}P(z_2, m, n, a, t)
\end{aligned}
\tag{6}
$$

with $P(z_i, m, n, a) := P(Z(t) = z_i, M(t) = m, N(t) = n, A \in [a + \mathrm{d}a) | \theta, \omega)$. Differential equations for arbitrary moments $\mathbb{E}[f(Z(t), M(t), N(t), A)]$ with $f$ as a polynomial can be computed by multiplying (6) with $f$ and summing or integrating over all possible values of $m$, $n$, $z_i$ and $a$, respectively (Zechner *et al*, 2012). In the following, we will denote by $\mathbf{s}_t = \{s(u) | 0 \leq u \leq t\}$ a complete sample path of the full system state between time zero and $t$ and introduce a corresponding path distribution $p(\mathbf{s}_t | \omega, \theta)$. The path distribution conditional on a particular initial state $S(0) = s_0$ is denoted by $p(\mathbf{s}_t | s_0, \omega, \theta)$.

### Conditional dynamics of transcription and translation

One major difficulty in inferring gene networks like the one in Fig 1 C is that they involve both very lowly and highly abundant components. This is why moment-based descriptions of the full system state $S(t)$ are of limited use for the time-series inference problem considered here as will be discussed later. On the other hand, approaches purely based on stochastic simulation become computationally expensive, since transcription and translation often involve thousands or even millions of events over the duration of a time-course experiment. In such cases, hybrid approaches can be beneficial, where only the lowly abundant components are described stochastically, whereas the remaining components are handled

using moment equations (Hasenauer *et al*, 2014). In the scenario considered here, for instance, the time evolution of the transcription rate $\mathbf{z}_t$ can be efficiently simulated using stochastic simulation since the number of times the promoter switches between states is comparably small. For a given $\mathbf{z}_t$, one could then calculate a corresponding set of conditional moments characterizing the dynamics of mRNA and protein. More technically, this can be understood by the fact that the path distribution over the total system state factorizes into $p(\mathbf{s}_t | \omega, \theta) = p(\mathbf{x}_t | \mathbf{z}_t, \omega)p(\mathbf{z}_t | \theta)$. Correspondingly, we can describe the dynamics over $X(t) = (M(t), N(t), A)$ as a conditional Markov process $X(t) | \mathbf{z}_t$, whose state probability distribution $P(m, n, a, t) := P(M(t) = m, N(t) = n, A \in [a + \mathrm{d}a) | \mathbf{z}_t)$ satisfies

$$
\begin{aligned}
\frac{d}{dt}P(m, n, a, t) &= z(t)P(m-1, n, a, t) + c_1(m+1)P(m+1, n, a, t)\\
&\quad + amP(m, n-1, a, t) + c_2(n+1)P(m, n+1, a, t)\\
&\quad - [z(t) + c_1 m + am + c_2 n]P(m, n, a, t),
\end{aligned}
\tag{7}
$$

where as we assume for the initial condition $P(m, n, a, t = 0) = P(M(0) = m, N(0) = n | Z(0) = z_0)p(a | \beta)$. For simplicity, we further consider the initial mRNA and protein copy numbers to be independent of the transcription rate such that $P(M(0) = m, N(0) = n | Z(0) = z_0) = P(M(0) = m, N(0) = n)$. In order to derive conditional moments, we multiply (7) with polynomials in $x$ and sum and integrate over all $m$, $n$, and $a$, respectively. Here, we consider moments of mRNA and protein up to order two, which can be fully described by the system of differential equations

$$
\begin{aligned}
\frac{\mathrm{d}}{\mathrm{d}t}\mathbb{E}[M(t)|\mathbf{z}_t] &= z(t) - \mathbb{E}[M(t)|\mathbf{z}_t]c_1\\
\frac{\mathrm{d}}{\mathrm{d}t}\mathbb{E}[N(t)|\mathbf{z}_t] &= \mathbb{E}[M(t)A|\mathbf{z}_t] - \mathbb{E}[N(t)|\mathbf{z}_t]c_2\\
\frac{\mathrm{d}}{\mathrm{d}t}\mathbb{E}\Big[M(t)^2|\mathbf{z}_t\Big] &= z(t) + 2\mathbb{E}[M(t)|\mathbf{z}_t]z(t) + \mathbb{E}[M(t)|\mathbf{z}_t]c_1\\
&\quad - 2\mathbb{E}\Big[M(t)^2|\mathbf{z}_t\Big]c_1\\
\frac{\mathrm{d}}{\mathrm{d}t}\mathbb{E}[M(t)N(t)|\mathbf{z}_t] &= \mathbb{E}[N(t)|\mathbf{z}_t]z(t) - \mathbb{E}[M(t)N(t)|\mathbf{z}_t]c_1\\
&\quad - \mathbb{E}[M(t)N(t)|\mathbf{z}_t]c_2 + \mathbb{E}\Big[M(t)^2 A|\mathbf{z}_t\Big]\\
\frac{\mathrm{d}}{\mathrm{d}t}\mathbb{E}[M(t)A|\mathbf{z}_t] &= \mathbb{E}[A|\mathbf{z}_t]z(t) - \mathbb{E}[M(t)A|\mathbf{z}_t]c_1\\
\frac{\mathrm{d}}{\mathrm{d}t}\mathbb{E}\Big[N(t)^2|\mathbf{z}_t\Big] &= \mathbb{E}[N(t)|\mathbf{z}_t]c_2 + \mathbb{E}[M(t)A|\mathbf{z}_t]\\
&\quad - 2\mathbb{E}\Big[N(t)^2|\mathbf{z}_t\Big]c_2 + 2\mathbb{E}[M(t)N(t)A|\mathbf{z}_t]\\
\frac{\mathrm{d}}{\mathrm{d}t}\mathbb{E}[N(t)A|\mathbf{z}_t] &= \mathbb{E}\Big[M(t)A^2|\mathbf{z}_t\Big] - \mathbb{E}[N(t)A|\mathbf{z}_t]c_2\\
\frac{\mathrm{d}}{\mathrm{d}t}\mathbb{E}\Big[M(t)^2 A|\mathbf{z}_t\Big] &= \mathbb{E}[A|\mathbf{z}_t]z(t) + 2\mathbb{E}[M(t)A|\mathbf{z}_t]z(t)\\
&\quad + \mathbb{E}[M(t)A|\mathbf{z}_t]c_1 - 2\mathbb{E}\Big[M(t)^2 A|\mathbf{z}_t\Big]c_1\\
\frac{\mathrm{d}}{\mathrm{d}t}\mathbb{E}[M(t)N(t)A|\mathbf{z}_t] &= \mathbb{E}[N(t)A|\mathbf{z}_t]z(t) - \mathbb{E}[M(t)N(t)A|\mathbf{z}_t]c_1\\
&\quad - \mathbb{E}[M(t)N(t)A|\mathbf{z}_t]c_2 + \mathbb{E}\Big[M(t)^2 A^2|\mathbf{z}_t\Big]\\
\frac{\mathrm{d}}{\mathrm{d}t}\mathbb{E}\Big[M(t)A^2|\mathbf{z}_t\Big] &= \mathbb{E}\Big[A^2|\mathbf{z}_t\Big]z(t) - \mathbb{E}\Big[M(t)A^2|\mathbf{z}_t\Big]c_1\\
\frac{\mathrm{d}}{\mathrm{d}t}\mathbb{E}\Big[M(t)^2 A^2|\mathbf{z}_t\Big] &= \mathbb{E}\Big[A^2|\mathbf{z}_t\Big]z(t) + 2\mathbb{E}\Big[M(t)A^2|\mathbf{z}_t\Big]z(t)\\
&\quad + \mathbb{E}\Big[M(t)A^2|\mathbf{z}_t\Big]c_1 - 2\mathbb{E}\Big[M(t)^2 A^2|\mathbf{z}_t\Big]c_1.
\end{aligned}
\tag{8}
$$

Note that (8) involves all first- and second-order moments, but also a few additional moments of order three and four, which are needed in order to obtain a closed set of differential equations.

### Statistical model of time-series reporter measurements

As detailed above, we analyzed quantitative single-cell time-lapse measurements of reporter expression for different Msn2-inducible promoters and Msn2 activation profiles. We denote by $t_1, \ldots, t_K$ the time points at which measurements were taken. Correspondingly, we define by $\mathbf{s}_{l:k}$ a complete sample path of the gene expression system between times $t_l$ and $t_k$. If $l = 0$, we refer to the state at time $t = 0$, which does not necessarily coincide with the first measurement time point $t_1$. The measurements—denoted by $Y_k$ for $k = 1, \ldots, K$—provide noisy information about the system state $S(t_k)$ according to a measurement density

$$Y_k | (S(t_k) = s_k) \sim p(\cdot | s_k).$$

We consider the measurement noise to be independent among time points such that

$$p(y_1, \ldots, y_K | s_1, \ldots, s_K) = \prod_{k=1}^{K} p(y_k | s_k). \tag{9}$$

In our particular case, the measurements correspond to the reporter abundance $N(t)$ corrupted by measurement noise such that

$$p(y_k | s_k) = p(y_k | x_k) = p(y_k | n_k).$$

For a given set of parameters $\{\theta, \omega\}$, the relation between a complete sample path $\mathbf{s}_{0:K}$ and the observed measurements is captured by a joint distribution

$$p(y_1, \ldots, y_K, \mathbf{s}_{0:K} | \omega, \theta) = p(\mathbf{s}_{0:K} | \omega, \theta) \prod_{k=1}^{K} p(y_k | s_k), \tag{10}$$

with $p(\mathbf{s}_{0:K} | \omega, \theta)$ as the distribution over complete sample paths $\mathbf{s}_{0:K}$. Correspondingly, the posterior distribution over $\mathbf{s}_{0:K}$ is proportional to (10), i.e.,

$$p(\mathbf{s}_{0:K} | y_1, \ldots, y_K, \omega, \theta) \propto p(\mathbf{s}_{0:K} | \omega, \theta) \prod_{k=1}^{K} p(y_k | s_k). \tag{11}$$

### Recursive Bayesian estimation

The posterior distribution (11) is generally intractable but several approximate techniques can be employed. Most of them rely on Bayesian filtering methods, which construct an approximation of (11) recursively over measurement time points. In those approaches, one exploits the fact that the posterior distribution at any measurement time $t_k$ can be written recursively as

$$\begin{aligned} p(\mathbf{s}_{0:k} | y_1, \ldots, y_k, \omega, \theta) &\propto p(y_k | s_k) p(\mathbf{s}_{k-1:k} | s_{k-1}, \omega, \theta) \\ &\quad p(\mathbf{s}_{0:k-1} | y_1, \ldots, y_{k-1}, \omega, \theta), \end{aligned} \tag{12}$$

with $p(\mathbf{s}_{0:k-1} | y_1, \ldots, y_{k-1}, \omega, \theta)$ as the posterior distribution at time $t_{k-1}$. In order to solve the Bayesian recursion between consecutive time steps, one can either employ analytical approximations, or

Monte Carlo methods. In a recent study, for instance, we have proposed normal and log-normal approximation of the Bayesian filtering problem, which rely on the time evolution of the first and second order moments of the gene network dynamics (Huang *et al*, 2016). While computationally efficient, the underlying continuous approximations may not be suitable for discrete and switch-like components, such as the transcription rate $Z(t)$ in our promoter model. Alternative approaches are mostly based on sequential Monte Carlo techniques, which approximate (11) using a sufficiently large number of Monte Carlo samples drawn by SSA. The main advantage of these techniques is that they are exact up to sampling variance but on their downside, suffer from limited scalability. In particular, forward simulation via SSA can become prohibitively slow, especially when RNAs and proteins are highly abundant. Therefore, they are currently not able to tackle large datasets like the one considered here. In the following, we will present a hybrid inference algorithm, which bypasses expensive SSA simulations of highly abundant species, making it sufficiently scalable to deal with datasets that span tens or even hundreds of thousands of single-cell trajectories.

### Hybrid sequential Monte Carlo

One strategy to improve the scalability of sequential Monte Carlo techniques is to analytically eliminate variables that are not of direct interest to a particular inference problem (Doucet *et al*, 2000; Zechner *et al*, 2014). In our case, for instance, we are specifically interested in the promoter switching dynamics and the corresponding transcription rate $Z(t)$. From this perspective, it would therefore suffice to calculate the marginal posterior distribution

$$p(\mathbf{z}_{0:K} | y_1, \ldots, y_K, \theta, \omega) \propto p(y_1, \ldots, y_K, \mathbf{z}_{0:K} | \theta, \omega) \tag{13}$$

in which the dynamics of $X(t)$ have been "integrated out". In order to perform this integration, we first realize that the joint distribution can be rewritten as

$$\begin{aligned} p(y_1, \ldots, y_K, \mathbf{s}_{0:K} | \omega, \theta) &= p(y_1, \ldots, y_K, \mathbf{x}_{0:K}, \mathbf{z}_{0:K} | \omega, \theta) \\ &= p(\mathbf{s}_{0:K} | \omega, \theta) \prod_{k=1}^{K} p(y_k | s_k) \\ &= p(\mathbf{x}_{0:K} | \mathbf{z}_{0:K}, \omega) p(\mathbf{z}_{0:K} | \theta) \prod_{k=1}^{K} p(y_k | x_k) \\ &= P(x_0) p(\mathbf{z}_{0:K} | \theta) \prod_{k=1}^{K} p(y_k | x_k) p(\mathbf{x}_{k-1:k} | x_{k-1}, \mathbf{z}_{k-1:k}, \omega) \end{aligned} \tag{14}$$

where we have made use of the identities $p(\mathbf{s}_{0:K} | \omega, \theta) = p(\mathbf{x}_{0:K} | \mathbf{z}_{0:K}, \omega) p(\mathbf{z}_{0:K} | \theta)$ and $p(\mathbf{x}_{0:K} | \mathbf{z}_{0:K}, \omega) = P(x_0) \prod_{k=1}^{K} p(\mathbf{x}_{k-1:k} | x_{k-1}, \mathbf{z}_{k-1:k}, \omega)$. Next, we integrate (14) over all subpaths $\mathbf{x}_{k-1:k} \setminus \{x_{k-1}, x_k\}$ such that only the values of $X(t)$ at the time points $t_0, \ldots, t_K$ remain in the model. Informally, this integration can be carried out by replacing the path distribution $p(\mathbf{x}_{k-1:k} | x_{k-1}, \mathbf{z}_{k-1:k}, \omega)$ by the state transition kernel $P(x_k | x_{k-1}, \mathbf{z}_{k-1:k}, \omega)$, i.e.,

$$\begin{aligned} p(y_1, \ldots, y_K, x_0, \ldots, x_K, \mathbf{z}_{0:K} | \omega, \theta) &= P(x_0) p(\mathbf{z}_{0:K} | \theta) \prod_{k=1}^{K} p(y_k | x_k) \\ &\quad P(x_k | x_{k-1}, \mathbf{z}_{k-1:k}, \omega). \end{aligned} \tag{15}$$

The marginalization over the remaining variables $x_0, \ldots, x_K$ then reduces to a summation

$$p(y_1,\ldots,y_K,\mathbf{z}_{0:K}|\theta,\omega) = \sum_{x_0}\ldots\sum_{x_K}p(y_1,\ldots,y_K,x_0,\ldots,x_K,\mathbf{z}_{0:K}|\omega,\theta). \quad (16)$$

Most conveniently, this summation can be solved iteratively, by first summing over $x_0$, subsequently over $x_1$ and so forth. The first summation yields

$$
\begin{aligned}
p(y_1,\ldots,y_K,x_1,\ldots,x_K,\mathbf{z}_{0:K}|\omega,\theta) &= \sum_{x_0}P(x_0)p(y_1|x_1)P(x_1|x_0,\mathbf{z}_{0:1},\omega)p(\mathbf{z}_{0:K}|\theta) \\
&\quad \times \prod_{k=2}^{K}p(y_k|x_k)P(x_k|x_{k-1},\mathbf{z}_{k-1:k},\omega) \\
&= p(y_1|x_1)P(x_1|\mathbf{z}_{0:1},\omega)p(\mathbf{z}_{0:K}|\theta) \\
&\quad \times \prod_{k=2}^{K}p(y_k|x_k)P(x_k|x_{k-1},\mathbf{z}_{k-1:k},\omega) \\
&= p(\mathbf{z}_{0:K}|\theta)P(x_1|y_1,\mathbf{z}_{0:1},\omega)p(y_1|\mathbf{z}_{0:1},\omega) \\
&\quad \times \prod_{k=2}^{K}p(y_k|x_k)P(x_k|x_{k-1},\mathbf{z}_{k-1:k},\omega), \quad (17)
\end{aligned}
$$

whereas the last step follows from the fact that $p(y_1|x_1)P(x_1|\mathbf{z}_{0:1},\omega) = P(x_1|y_1,\mathbf{z}_{0:1},\omega)p(y_1|\mathbf{z}_{0:1},\omega)$ via Bayes' rule. Repeating the same procedure for $x_1$ yields

$$
\begin{aligned}
&p(y_1,\ldots,y_K,x_2,\ldots,x_K,\mathbf{z}_{0:K}|\omega,\theta) \\
&= \sum_{x_1}p(\mathbf{z}_{0:K}|\theta)P(x_1|y_1,\mathbf{z}_{0:1},\omega)p(y_1|\mathbf{z}_{0:1},\omega) \\
&\quad \times \prod_{k=2}^{K}p(y_k|x_k)P(x_k|x_{k-1},\mathbf{z}_{k-1:k},\omega) \\
&= p(\mathbf{z}_{0:K}|\theta)p(y_2|x_2)\sum_{x_1}P(x_2|x_1,\mathbf{z}_{1:2},\omega)P(y_1,\mathbf{z}_{0:1},\omega)p(y_1|\mathbf{z}_{0:1},\omega) \\
&\quad \times \prod_{k=3}^{K}p(y_k|x_k)P(x_k|x_{k-1},\mathbf{z}_{k-1:k},\omega) \\
&= p(\mathbf{z}_{0:K}|\theta)p(y_2|x_2)P(x_2|y_1,\mathbf{z}_{0:2},\omega)p(y_1|\mathbf{z}_{0:1},\omega) \\
&\quad \times \prod_{k=3}^{K}p(y_k|x_k)P(x_k|x_{k-1},\mathbf{z}_{k-1:k},\omega) \\
&= p(\mathbf{z}_{0:K}|\theta)P(x_2|y_2,y_1,\mathbf{z}_{0:2},\omega)p(y_2|y_1,\mathbf{z}_{0:2},\omega)p(y_1|\mathbf{z}_{0:1},\omega) \\
&\quad \times \prod_{k=3}^{K}p(y_k|x_k)P(x_k|x_{k-1},\mathbf{z}_{k-1:k},\omega).
\end{aligned} \quad (18)
$$

Continuing the above procedure for $x_2,\ldots,x_K$ finally leads to.

$$
\begin{aligned}
p(y_1,\ldots,y_K,\mathbf{z}_{0:K}|\omega,\theta) &= p(\mathbf{z}_{0:K}|\theta)p(y_1|\mathbf{z}_{0:1},\omega) \\
&\quad \prod_{k=2}^{K}p(y_k|y_{k-1},\ldots,y_1,\mathbf{z}_{0:k},\omega).
\end{aligned} \quad (19)
$$

Therefore, the marginal posterior distribution over the transcription dynamics $\mathbf{z}_{0:K}$ is proportional to (19), which can also be expressed recursively as.

$$
\begin{aligned}
p(\mathbf{z}_{0:K}|y_1,\ldots,y_K,\omega,\theta) &\propto p(y_1,\ldots,y_K,\mathbf{z}_{0:K}|\omega,\theta) \\
&\propto p(y_K|y_{K-1},\ldots,y_1,\mathbf{z}_{0:K},\omega) \\
&\quad p(\mathbf{z}_{K-1:K}|z_{K-1},\theta) \\
&\quad p(\mathbf{z}_{0:K-1}|y_1,\ldots,y_{K-1},\omega,\theta).
\end{aligned} \quad (20)
$$

Importantly, using equation (20) we can perform a sequential Monte Carlo algorithm on a significantly reduced sampling space, where only the transcription dynamics $\mathbf{z}_{0:K}$ have to be simulated explicitly. However, in order to perform this algorithm, we need to be able to calculate the marginal likelihood terms $p(y_k|y_{k-1},\ldots,y_1,\mathbf{z}_{0:k},\omega)$, which are given by

$$
\begin{aligned}
p(y_k|y_{k-1},\ldots,y_1,\mathbf{z}_{0:k},\omega) &= \sum_{x_k}p(y_k|x_k)P(x_k|y_{k-1},\ldots,y_1,\mathbf{z}_{0:k},\omega) \\
&= \sum_{x_k}p(y_k|x_k)\sum_{x_{k-1}}P(x_k|x_{k-1},\mathbf{z}_{k-1:k},\omega) \\
&\quad P(x_{k-1}|y_{k-1},\ldots,y_1,\mathbf{z}_{0:k-1},\omega). \quad (21)
\end{aligned}
$$

The two sums in (21) are gerally intractable, but analytical solutions exist if the measurement likelihood function $p(y_k|x_k)$ and the state transition kernel $P(x_k|x_{k-1},\mathbf{z}_{k-1:k},\omega)$ belong to certain classes of distributions. This is the case, for instance, if both are Gaussian. However, this is likely not a good assumption in the scenario considered here, since both the measurement and state distributions are generally positive and asymmetric. As it turns out, however, equation (11) has an analytical solution also if both $p(y_k|x_k)$ and $P(x_k|x_{k-1},\mathbf{z}_{k-1:k},\omega)$ are log-normally distributed. Log-normal distributions have been used previously to model measurement noise in time-lapse fluorescence data (Zechner *et al*, 2014) and gene product distributions (Taniguchi *et al*, 2010). We therefore assume[3]

$$
\begin{aligned}
Y_k|(N(t_k) = n) &\sim \mathcal{LN}(\log(n),\eta^2) \\
X(t)|\mathbf{z}_t &\sim \mathcal{LN}(\mu(t),\Sigma(t)),
\end{aligned} \quad (22)
$$

where $\eta^2$ corresponds to the strength of the measurement noise and $\mu(t)\in\mathbb{R}^3$ and $\Sigma(t)\in\mathbb{R}^{3\times3}$ characterize the distribution over $X(t) = (M(t),N(t),A)$ conditionally on a particular realization of $\mathbf{z}_t$. More precisely, $\mu(t)$ and $\Sigma(t)$ are the mean and covariance of $\log(X(t))$ and we therefore refer to them as logarithmic moments in the following.

Now, assuming that the posterior distribution over $X(t)$ is log-normally distributed at time $t_{k-1}$,

$$p(x_{k-1}|y_{k-1},\ldots,y_1,\mathbf{z}_{0:k-1},\omega) \approx \mathcal{LN}(x_{k-1}|\mu(t_{k-1}),\Sigma(t_{k-1})), \quad (23)$$

it will—based on our assumption—remain log-normal upon applying the state transition kernel, i.e.,

$$
\begin{aligned}
p(x_k|y_{k-1},\ldots,y_1,\mathbf{z}_{0:k},\omega) &\approx \int p(x_k|x_{k-1},\mathbf{z}_{k-1:k},\omega) \\
&\quad \mathcal{LN}(x_{k-1}|\mu(t_{k-1}),\Sigma(t_{k-1}))\mathrm{d}x_{k-1} \\
&\approx \mathcal{LN}(x_k|\mu(t_k),\Sigma(t_k)),
\end{aligned} \quad (24)
$$

where the sum has now been replaced by an integral. In order to calculate the logarithmic moments $\mu(t_k)$ and $\Sigma(t_k)$ for a given $\mu(t_{k-1})$ and $\Sigma(t_{k-1})$, one first has to calculate all moments that enter equation (8) from the log-normal distribution, propagate those forward in time until $t_k$ using (8), and subsequently convert them back into the logarithmic domain to obtain $\mu(t_k)$ and $\Sigma(t_k)$. For instance, the relationship between logarithmic and standard moments of order one and two is given by

$$
\begin{aligned}
\mathbb{E}[X_i(t)] &= e^{\mu_i(t)+\frac{1}{2}\Sigma_{ii}(t)} \\
\mathbb{E}[X_i(t)X_j(t)] &= e^{\mu_i(t)+\mu_j(t)+\frac{1}{2}\left(\Sigma_{ii}(t)+2\Sigma_{ij}(t)+\Sigma_{jj}(t)\right)}.
\end{aligned} \quad (25)
$$

In order to determine the posterior distribution at the next measurement time $t_k$, we multiply (24) with the log-normal measurement density such that

$$p(x_k|y_k,\ldots,y_1,\mathbf{z}_{0:k},\omega) \propto p(y_k|x_k)p(x_k|y_{k-1},\ldots,y_1,\mathbf{z}_{0:k},\omega)$$
$$= \mathcal{LN}(y_k|\log(n_k),\eta^2) \times \mathcal{LN}(x_k|\mu(t_k),\Sigma(t_k)), \quad (26)$$

with $n_k = N(t_k)$ as the protein abundance at time $t_k$. One can show that the product of the two log-normal distributions in (26) is again proportional to a log-normal distribution such that

$$p(x_k|y_k,\ldots,y_1,\mathbf{z}_{0:k},\omega) = \mathcal{LN}(x_k|\mu^+(t_k),\Sigma^+(t_k)), \quad (27)$$

with

$$\Sigma^+(t_k) = \left[\frac{1}{\eta^2}ww^T + \Sigma(t_k)^{-1}\right]^{-1} \quad (28)$$

$$\mu^+(t_k) = \Sigma^+(t_k)\left[\frac{1}{\eta^2}\log(y_k)w + \Sigma(t_k)^{-1}\mu(t_k)\right], \quad (29)$$

and $w = (0,1,0)^T$ as a vector that reflects the fact that from $X(t) = (M(t),N(t),A)$, the second component (i.e., the protein abundance) is measured experimentally.

For the likelihood term $p(y_k|y_{k-1},\ldots,y_1,\mathbf{z}_{0:k},\omega)$ we obtain

$$p(y_k|y_{k-1},\ldots,y_1,z_{0:k},\omega) = \int p(y_k|x_k)p(x_k|y_{k-1},\ldots,y_1,z_{0:k},\omega)dx_k$$
$$= \int \mathcal{LN}(y_k|\log(n_k),\eta^2)p(n_k|y_{k-1},\ldots,y_1,z_{0:k},\omega)dn_k$$
$$= \int \mathcal{LN}(y_k|\log(n_k),\eta^2)\mathcal{LN}(n_k|\mu_2(t_k),\Sigma_{22}(t_k))dn_k, \quad (30)$$

where the last line follows from the fact that each dimension $i$ of a multivariate log-normal distribution with logarithmic moments $\mu$ and $\Sigma$ is marginally log-normal with parameters $\mu_i$ and $\Sigma_{ii}$. This integral can be solved in closed form such that we obtain for the logarithm of the marginal likelihood function

$$\log p(y_k|y_{k-1},\ldots,y_1,\mathbf{z}_{0:k},\omega) =$$
$$-\frac{1}{2}\left[\frac{(\log y_k - \mu_2(t_k))^2}{\eta^2 + \Sigma_{22}(t_k)} - \log\left(\frac{1}{\eta^2} + \Sigma_{22}(t_k)^{-1}\right)\right.$$
$$\left. -\log(\Sigma_{22}(t_k)) - \log(\eta^2) - \log(y_k) + const.\right] \quad (31)$$

Together, equations (8), (28), (29), and (31) define a recursive Bayesian filter, which allows us to eliminate the components $X(t)$ from the inference problem. As mentioned above, the remaining component $Z(t)$ can then be inferred efficiently using a conventional sequential importance sampler. To this end, we define a set of $J$ particles, each of them consisting of a path $\mathbf{z}^{(i)}$, a set of logarithmic moments $\mu^{(i)}$ and $\Sigma^{(i)}$ as well as a particle probability $p^{(i)}$. This set of particles serves as a finite sample approximation of the posterior distribution at each iteration $k$. At the $k^{\text{th}}$ time step, $J$ new particles are drawn randomly according to the particle probabilities $p^{(i)}$. For each particle $i$, the path $\mathbf{z}^{(i)}$ is first extended to the next measurement $t_{k+1}$ using SSA. The new probability of this particle is then determined by first propagating the corresponding logarithmic moments until $t_{k+1}$ using equation (8) and then evaluating equation (31). The particle probabilities are then normalized across the $J$ particles such that they sum up to one. Subsequently, $\mu^{(i)}$ and $\Sigma^{(i)}$ are updated using (28) and (29) and the algorithm proceeds with the next iteration. At the final time $t_K$, the paths $\mathbf{z}^{(i)}$ associated with the particles represent samples from the desired marginal posterior distribution, which can be used for further analysis.

### Quantitative characterization of promoter dynamics

The inference algorithm described above allows as to compute an arbitrary number of samples $\mathbf{z}_{0:K}^{(i)}$ from the desired posterior distribution. In order to compare the dynamics of the different promoters under various experimental conditions, we extracted a number of features from these samples that characterize the transcriptional response for each individual cell. More technically, these features can be defined as functionals that map a random path $\mathbf{z}_{0:K}^{(i)}$ to a real or discrete number. This functional can then be averaged with respect to the posterior distribution associated with a particular cell, i.e.,

$$\mathbb{E}[f(\mathbf{z}_{0:K})|y_1,\ldots,y_K] \approx \frac{1}{J}\sum_{j=1}^{J}f\left(\mathbf{z}_{0:K}^{(j)}\right), \quad (32)$$

with $y_1,\ldots,y_K$ as the measurements of this cell and $\mathbf{z}_{0:K}^{(j)}$ as samples from the posterior distribution obtained from the inference method. The following list summarizes the different features that were used in this study.

- Responding/non-responding. A cell is considered a responder if it resided in a state of significant transcriptional activity for at least 2 min. To this end, we defined a functional $r(\mathbf{z}_{0:K}) \in \{0,1\}$, which is one only if this criterion is met. We define a transcriptionally significant state as one that has a transcription rate of at least 20% of the maximum transcription rate taken over all 50 min pulse conditions. Depending on the promoter and condition, this could encompass one, two, or none of the promoter states. We then estimated the response probability $p_a = \mathbb{E}[r(\mathbf{z}_{0:K})|y_1,\ldots,y_K]$ for each cell by averaging over all the individual samples paths $\mathbf{z}_{0:K}$ obtained from the sequential Monte Carlo algorithm. A cell was then classified as a responder if $p_a > 0.99$. Subsequently, we calculated the percentage of responders for each promoter and condition.
- Time to activate. For all responding cells, we calculated the posterior expectation of the time it took until the cell switched into a transcriptionally significant state, i.e., $\mathbb{E}[\tau_S(\mathbf{z}_{0:K})|y_1,\ldots,y_K]$ with $\tau_S(\mathbf{z}_{0:K}) \in \mathbb{R}^+$ as a functional that measures the time until the first transition into a responsive state happened. Paths for which the promoter was in a responsive state for less than two minutes were excluded from this expectation. We further calculated the mean and variance of the time until activation over all cells in an experiment.
- Total time active. Analogously to the time to activate, we quantified the total time the promoter was active, i.e., $\mathbb{E}[\tau_A(\mathbf{z}_{0:K})|y_1,\ldots,y_K]$ with $\tau_A(\mathbf{z}_{0:K}) \in \mathbb{R}^+$ as a functional that extracts the total time the promoter spent in any of the active states.
- Time spent in state $i$. We calculated the total time the promoter spent in any of the three states, i.e., $\mathbb{E}[\tau_i(\mathbf{z}_{0:K})|y_1,\ldots,y_K]$ with $\tau_i(\mathbf{z}_{0:K}) \in \mathbb{R}^+$.
- Maximum transcription. We calculated the maximum transcription rate that the promoter achieved during a time-course experiment. In particular, we computed the expected transcription rate for each cell $\lambda(t) = \mathbb{E}[Z(t)|y_1,\ldots,y_K]$ and subsequently the corre-

sponding population average $\langle\lambda(t)\rangle$, whereas only cells that were classified as responders were considered. We then determined the maximum of this average, i.e, $\lambda_{max} = \max_t \langle\lambda(t)\rangle$.

- Time to maximum transcription. Next to the maximum transcription, we also determined the time when this maximum was achieved, i.e., $\tau_{max} = \arg\max_t \langle\lambda(t)\rangle$.
- Transcriptional output. To quantify the amount of transcription along a whole time course, we calculated the integral over the inferred transcription rates, i.e., $o = \mathbb{E}\left[\int_0^t Z(s)\mathrm{d}s | y_1, \ldots, y_K\right]$.

### Evaluation of the inference method using synthetic data

In order to study the accuracy of the proposed inference method, we tested it using artificially generated data. In particular, we considered two differently parameterized versions of the stochastic model in Fig 1C. The first one resembled a fast promoter like *DCS2* or *HXK1* whereas the second one had slow and switch-like promoter activation kinetics like *SIP18* or *TKL1*. In particular, the parameters of the system were chosen to be $\gamma = 0.05$, $q_{10} = 0.055$, $q_{12} = 0.001\kappa$, $q_{21} = 0.004\kappa$, $z_1 = 0.0035$, $z_2 = 0.728$, $c_1 = 0.0013$, $c_2 = 1.67e-5$, $\langle A \rangle = 0.1$, $CV[A] = 0.02$, whereas $\kappa = \{1, 10\}$ for the slow and fast promoter model, respectively. All rate parameters are given in units $s^{-1}$.

For each promoter, we generated 30 single-cell trajectories between time zero and $t_K = 150$min using SSA and sampled the protein abundance at 55 equidistant time points $t_1, \ldots, t_K$. For the Msn2 activation function $u(t)$, we used the experimentally determined profile for a single-pulse experiment (75% Msn2 induction level, 40min duration). The measurements were then simulated from a log-normal measurement density $\mathcal{LN}(y_k | \log(n_k), \eta^2)$, with $n_k$ as the protein copy number at time $t_k$ and $\eta$ as the logarithmic standard deviation of this density. For this study, we set $\eta = 0.05$.

We applied the hybrid sequential Monte Carlo algorithm to reconstruct the promoter dynamics and compared it with the true realization. In particular, we analyzed three of the path functionals described in Section "Quantitative characterization of promoter dynamics": total time active, time to activate and transcriptional output. We estimated posterior expectations of these functionals using $J = 400$ Monte Carlo samples and analyzed how they compared with the true values extracted from the exact sample paths $\mathbf{z}_{0:K}$. We first assumed perfect knowledge of all process parameters. The top panels in Fig EV2A and B show the inferred values plotted against the ground truth. For all three features, we found a linear relationship with a slope $k$ close to one. The corresponding $R^2$ indicates the reconstruction accuracy of the inference method. For the slowly switching promoter, we found $R^2$ values close to one, indicating very high accuracy. For the fast-switching promoter, the inference results become slightly less accurate because individual switching events are more difficult to infer from the relatively slow reporter dynamics. We furthermore analyzed the robustness of the method with respect to parameter mismatch. To this end, we randomly perturbed all of the parameters using a log-normal distribution $\mathcal{LN}(\log(b), 0.1^2)$ with $b$ as the underlying true value. Note that the random parameter perturbation was performed for each of the considered trajectories separately. In case of poor robustness, we would thus expect a significantly reduced correlation between the true and inferred values. However, we found for all three features that both the $R^2$ and slope $k$ changed only marginally indicating a relatively high robustness of the method. This is an

important feature in practical scenarios where knowledge about process parameters is generally imperfect.

### Statistical analysis of Msn2-dependent gene expression

In the following, we provide details on the statistical analysis of Msn2-dependent gene expression as shown in the main text. In this case, the function $u(t)$ corresponds to the nuclear Msn2 level that was measured experimentally for each condition (Fig EV1). In combination with the measured YFP time series, this allowed us to infer the input–output relationship of different promoters under different experimental conditions using the recursive inference method described in Section "Hybrid sequential Monte Carlo". However, before this method could be applied, the stochastic model from Fig 1C had to be parameterized. For this purpose, we used a portion of the experimental single-cell trajectories to infer the kinetic parameters of the model (Section "Statistical inference of kinetic parameters"). Subsequently, we reconstructed the transcription dynamics of each promoter and condition as described in Section "Statistical inference of transcription dynamics".

### Statistical inference of kinetic parameters

In order to parameterize the stochastic gene expression model for different promoters and experimental conditions, we used an established moment-based inference method (Zechner *et al*, 2012). This method uses a Markov chain Monte Carlo sampler to match the first and second order moments of the stochastic gene network to the experimentally determined ones. For detailed information on this approach, the reader shall refer to (Zechner *et al*, 2012).

For each promoter, we first estimated the total set of parameters $\omega$ and $\theta$ using the single-pulse experiments with maximum level and duration (100% Msn2, 50 min). Since the promoter switching dynamics can be concentration- and pulse length-dependent, we re-estimated the promoter parameters $\theta$ for all other conditions, while keeping $\omega$ fixed at the previously inferred values. The kinetics of the same gene expression system have been previously quantified using a deterministic model (Hansen & O'Shea, 2013). We incorporated this additional information in the form of prior distributions over some of the kinetic parameters. In particular, we considered Gamma prior distributions $p(c_1) = \Gamma(20, 20/1.3e-3s^{-1})$ and $p(\langle A \rangle) = \Gamma(20, 20/0.05s^{-1})$ for the mRNA degradation and average protein translation rates, respectively. Additionally, the protein degradation rate was fixed to $c_2 = 1.67e-5s^{-1}$. For the switching parameters $q_{ij}$ and the transcription rates $z_1$ and $z_2$, we used prior distributions $p(\cdot) = \Gamma(1, 1/30s^{-1})$. To infer the parameters, we applied a Metropolis-Hastings sampler with log-normal proposal distributions to generate $2e4$ samples from which we extracted maximum a posterior (MAP) estimates of the model parameters.

### Statistical inference of transcription dynamics

Using the calibrated models, we inferred the transcription and promoter switching dynamics using the hybrid sequential Monte Carlo inference scheme from Section "Hybrid sequential Monte Carlo". Based on our previous study (Zechner *et al*, 2014), which uses a similar data processing and calibration pipeline, we set the measurement noise parameter to $\eta = 0.15$ corresponding to an expected relative variation of roughly 15 percent. For each condition and promoter, we processed each individual cell using $J = 400$ particles.

From the resulting particles, we estimated the promoter features as summarized in Section "Quantitative characterization of promoter dynamics". We note that in some circumstances, the hybrid SMC algorithm can become numerically unstable. For instance, this may be the case in the presence of outliers, where two consecutive data points are very far away from each other. All cells that led to unstable results were excluded from our analyses. The ratio of excluded cells was fairly small for most promoters and conditions (i.e., for around 90% of the 270 experiments less than 15% of trajectories were excluded). For a small fraction of around 3% of the experiments, between 30 and 50% of the trajectories had to be dismissed. However, all these experiments correspond to promoters and conditions were gene expression signals were very low and close to background. Therefore, the exclusion of trajectories should affect our analyses to no significant extent. Moreover, we performed a quantitative analysis, which shows that the exclusion of trajectories did not strongly affect the statistical properties of the gene expression levels for individual promoters and conditions. The corresponding analysis can be found in the provided GitHub repository.

As indicated in the main text, the overall analysis pipeline depends on random number generation (e.g., splitting of data between model calibration and reconstruction, MCMC sampling during parameter estimation, sequential Monte Carlo inference), and therefore, the inferred transcriptional features exhibit a certain degree of variability between repeated runs of the analysis. To quantify this uncertainty, we performed the overall analysis five times and calculated averages and standard errors of the resulting transcriptional features. Note that certain transcriptional features are defined only for responding cells (e.g., time to activate). For conditions that contain only a small number responding cells, it can happen that in some of the repeated runs, no responders are detected, which leaves those transcriptional features undefined. In these cases, averages and standard errors were calculated over all runs for which the number of responders was non-zero.

### Toy model of interval-dependent promoter memory

For the simulations shown in Fig EV4, we considered a simple promoter model described by a reaction network.

$$
\begin{aligned}
P_0 &\underset{c_2}{\overset{c_1 u(t)}{\rightleftharpoons}} P_1 \\
P_1 &\underset{c_4}{\overset{c_3 I_1(t)}{\rightleftharpoons}} P_2 \\
P_0 &\underset{c_6}{\overset{c_5 I_2(t)}{\rightleftharpoons}} P_3 \\
P_1 &\overset{c_7}{\rightarrow} P_1 + I_1 \\
P_0 + I_1 &\overset{c_8}{\rightarrow} P_0 + I_2
\end{aligned}
\tag{33}
$$

with $u(t)$ as the experimentally measured nuclear Msn2 abundance. Transcription takes place with rate $z$ when the promoter is in state $P_2$. The parameters used for simulation were chosen to be $c_1 = 0.02$, $c_2 = 0.06$, $c_3 = 0.003$, $c_4 = 0.02$, $c_5 = 0.0006$, $c_6 = 0.001$, $c_7 = 0.9$, $c_8 = 7e - 6$, and $z = 0.6$ in units $s^{-1}$.

### Toy model of a complex, context-dependent promoter

We performed simulations of a four-state promoter model with nonlinear Msn2-dependent switching rates. In summary, theodel is described by a reaction network.

$$
\begin{aligned}
P_0 &\underset{c_2}{\overset{c_1(t)}{\rightleftharpoons}} P_1 \\
P_1 &\underset{c_4(t)}{\overset{c_3(t)}{\rightleftharpoons}} P_2 \\
P_2 &\underset{}{\overset{c_5(t)}{\rightleftharpoons}} P_3 \\
P_3 &\overset{c_6}{\rightarrow} P_0 \\
P_1 &\overset{z_1}{\rightarrow} P_1 + M \\
P_3 &\overset{z_3}{\rightarrow} P_3 + M
\end{aligned}
\tag{34}
$$

with

$$
c_1(t) = \gamma_1 u(t)
\tag{35}
$$

$$
c_3(t) = \gamma_3 \left( 1 - \frac{u(t)^{n_3}}{V_3^{n_3} + u(t)^{n_3}} \right)
\tag{36}
$$

$$
c_4(t) = \gamma_4 \frac{u(t)^{n_4}}{V_4^{n_4} + u(t)^{n_4}}
\tag{37}
$$

$$
c_5(t) = \gamma_5 \frac{u(t)^{n_5}}{V_5^{n_5} + u(t)^{n_5}}
\tag{38}
$$

and $\gamma_1 = c_2 = \gamma_3 = \gamma_4 = c_6 = 0.01/s$, $\lambda_5 = 0.1/s$, $n_3 = 6$, $n_4 = 2$, $n_5 = 3$, $V_3 = 0.5$, $V_4 = 0.001$, $V_5 = 1.2$. The symbol $u(t)$ denotes the time-varying Msn2 input in arbitrary units and the species $M$ in (34) corresponds to mRNA. The two transcription rates $z_1$ and $z_3$ are considered to be non-zero but their specific value is irrelevant for the purpose of this analysis. The promoter can be described by a forward equation

$$
\frac{d}{dt}
\begin{pmatrix}
P_1(t) \\
P_2(t) \\
P_3(t) \\
P_4(t)
\end{pmatrix}
=
\underbrace{
\begin{pmatrix}
-c_1(t) & c_2 & 0 & c_6 \\
c_1(t) & -(c_2 + c_3(t) + c_5(t)) & c_4(t) & 0 \\
0 & c_3(t) & -c_4(t) & 0 \\
0 & c_5(t) & 0 & -c_6
\end{pmatrix}
}_{Q(t)}
\begin{pmatrix}
P_1(t) \\
P_2(t) \\
P_3(t) \\
P_4(t)
\end{pmatrix}
\tag{39}
$$

with generator $Q(t)$. From the solution of the forward equation, we can directly calculate the expected number of state transitions by multiplying the entries of $Q(t)$ with the respective state probabilities and integrating over time. In particular, we calculated

$$
H(t) = \int_0^t Q(s) P(s) ds,
\tag{40}
$$

with matrix $P(t)$ defined as

$$
P(t) =
\begin{pmatrix}
P_0(t) & 0 & 0 & 0 \\
0 & P_1(t) & 0 & 0 \\
0 & 0 & P_2(t) & 0 \\
0 & 0 & 0 & P_3(t)
\end{pmatrix}.
\tag{41}
$$

The resulting matrix $H(t)$ counts the expected number of transitions between all states between time zero and $t$. The diagonal elements of the matrix correspond to the (negative) total number of transitions from one state to *any* other state. In Fig 4B in the main text, we show the matrix $H(t)$ for different dynamical inputs, whereas the diagonal elements were set to zero for clarity.

## Data availability

All source data files and software code supporting this manuscript are available from the following resources:

- Unprocessed and processed source data: Zenodo, http://doi.org/10.5281/zenodo.2755026, (http://zenodo.org/record/2755026)
- Computer code: GitHub, https://github.com/zechnerlab/PromoterManifest/

**Expanded View** for this article is available online.

## Acknowledgements

ASH acknowledges support from the Howard Hughes Medical Institute (to Erin K. O'Shea), the Siebel Stem Cell Foundation (post-doctoral fellowship), and the National Institutes of Health (R00GM130896 and DP2GM140938) during parts of this work. CZ acknowledges support from the Max Planck Society and the MPI-CBG. We thank Nan Hao, Nadine Vastenhouw, Stephan Grill, Andre Nadler, Carl Modes, Alf Honigmann, Pavel Tomancak, and Lorenzo Duso for insightful comments on the manuscript. Open access funding enabled and organized by ProjektDEAL.

## Author contributions

Study conception, data analysis, figures, and manuscript drafting and editing: ASH and CZ. Bayesian Inference method: CZ.

## Conflict of interest

The authors declare that they have no conflict of interest.

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
