## [Review Process File · Molecular Systems Biology]

Promoters adopt distinct dynamic manifestations depending on transcription factor context

Christoph Zechner and Anders Hansen

DOI: [10.15252/msb.20209821](https://doi.org/10.15252/msb.20209821)

Corresponding author(s): Christoph Zechner (zechner@mpi-cbg.de) , Anders Hansen (ashansen@mit.edu)

Review Timeline:	Editor Correspondence:	6th June 20
	Submission Date:	29th Jun 20
	Editorial Decision:	20th Jul 20
	Revision Received:	15th Aug 20
	Accepted:	25th Aug 20

Editor: Thomas Lemberger (MSB)

Transaction Report:

Please note that this manuscript was previously reviewed at another journal. Since the original reviews are not subject to The EMBO Journal's transparent review process policy, these initial reports and author response to them cannot be published here.

Dear Christoph,

Thank you again for your patience. We have now had the chance to discuss the study at our editorial meeting. We would like to invite you to formally submit the paper to the journal via the submission site (msb.msubmit.net). We would kindly ask you to address the following points:

1. It might be useful to reiterate in the discussion the difference between this study (ie. Intrinsic dynamics of the same promoter can change depending on stimulus patters and context) and previous studies (such as the ones mentioned by reviewer #2) that showed that one promoter can filter or discriminate stimuli patterns). This may help some readers to better understand the main message of your paper.

2. I would also suggest you to consider including some further discussion about the foundation of the concept of 'promoter state', which is the framework used to describe promoter dynamics. The fundamental assumption underlying this study (and irrespectively of any modeling details) is that promoter dynamics can be captured by representing them as adopting *discrete* states and characterizing the switching dynamics between such states. I wonder whether the phenomenon observed in this study, ie. different context-dependent 'manifestations', could also reveal that this discretization of promoter dynamics into 'states' is too coarse grained to capture the full complexity underlying the dynamical properties of promoters. Could your observations also be interpreted as suggesting that a promoter 'state' (or some of its features) can in fact not be reduced to discrete states but can only be described as a continuous state (something like a non-linear function)? The justification of discrete states vs continuous states representation will eventually be motivated by knowing how well they represent the molecular mechanisms that determine promoter dynamics. Is it possible that the observation of context-dependent 'manifestations' reveals that at least some rate-limiting biochemical events are better represented as continuous functions rather than discrete states? I realize that these questions were not mentioned by any of the four referees but I thought that it could be insightful to have some further thoughts on these issues as it may reveal some path forward in terms of conceptualizing and modeling transcription and finding the molecular explanations of the observed context-dependent promoter dynamical properties. Would this make sense?

4. The issue related to whether the system is 'feedback-less' is best addressed by some toning down and clarifying that formal exclusion of any feedback is difficult if not impossible.

Once the manuscript has been submitted, we will NOT send the paper for a new round of review and will aim at making a final decision as fast as possible.

RESPONSE TO CORRESPONDENCE

Please find our responses to comments below in a point-by-point format.

Thomas Lemberger comments:

1. It might be useful to reiterate in the discussion the difference between this study (ie. Intrinsic dynamics of the same promoter can change depending on stimulus patters and context) and previous studies (such as the ones mentioned by reviewer #2) that showed that one promoter can filter or discriminate stimuli patterns). This may help some readers to better understand the main message of your paper.

Response: We agree, and recognize that we should have more incisively discussed that while prior studies have demonstrated distinct promoter classes that can discriminate stimuli patterns, prior studies (including by us) had assumed that promoter class was a fixed property. Here we show that promoter class is not fixed, but can be completely changed depending on TF context. We lay out this distinction with special emphasis in the text pertaining to Fig. 2 and in the Discussion, and have copied this below.

Fig. 2: *“While it is well-known that promoters fall into distinct classes (Haberle and Stark, 2018; Hansen and O’Shea, 2013; 2015; Hao and O’Shea, 2012; King et al., 2020; Sharon et a., 2012; Suter et al., 2011; Stavera et al., 2009), what we show here is that the same promoter can switch from one class to another depending on context. To explain this phenomenon, we introduce the concept of “context-dependent manifestations”. Operationally, we define a context-dependent manifestation of a promoter as a situation where the same promoter exhibits qualitatively distinct kinetic behaviors under different input contexts.”*

Discussion: *“Previously, a large number of prior studies have shown that promoter fall into distinct classes (e.g. fast vs. slow; low vs. high threshold) and that different promoters decode dynamic stimuli differently (Haberle and Stark, 2018; Hansen and O’Shea, 2013; 2015; Hao and O’Shea, 2012; King et al., 2020; Sharon et a., 2012; Suter et al., 2011; Stavera et al., 2009). For example, a slow promoter may filter out a brief and transient stimulus (Purvis and Lahav, 2013). However, promoter class was assumed to be a fixed property.*

Here, we show that promoters can switch between classes depending on context. We show that even under these relatively simple conditions, the same promoter can exhibit context-dependent scaling and induction behaviors. To describe this observation, we introduce the concept of context-dependent “manifestations”. The underlying number of molecular states of a promoter is potentially enormous: when we measure a dose-response, we likely observe only certain rate-limiting regimes or manifestations of the system. What we show here is that the particular observed rate-limiting manifestation is highly context-dependent and very distinct quantitative behaviors can be observed under different contexts – even in systems that are seemingly simple.”

We hope these new text segments more clearly explain this difference, but we would be happy to take your recommendation on additional ways to clearly lay out this point or on way to modify these text segments. Clearly communicating this difference between prior studies (fixed promoter classes) and this study (promoter class switching depending on context) is foundational to communicating the conceptual advance of this study. We also emphasize this point now in a sentence in the abstract.

2. I would also suggest you to consider potentially including some further discussion about the foundation of the concept of 'promoter state', which is the framework used to describe promoter dynamics. The fundamental assumption underlying this study (and irrespectively of any modeling details) is that promoter dynamics can be captured by representing them as adopting *discrete* states and characterizing the switching dynamics between such states. I wonder whether the phenomenon observed in this study, ie. different context-dependent 'manifestations', could also reveal that this discretization of promoter dynamics into 'states' is too coarse grained to capture the full complexity underlying the dynamical properties of promoters. Could your observations also be interpreted as suggesting that a promoter 'state' (or some of its features) can in fact not be reduced to discrete states but can only be described as a continuous state (something like a non-linear function)? The justification of discrete states vs continuous states representation will eventually be motivated by knowing how well they represent the molecular mechanisms that determine promoter dynamics. Is it possible that the observation of context-dependent 'manifestations' reveals that at least some rate-limiting biochemical events are better represented as continuous functions rather than discrete states? I realize that these questions were not mentioned by any of the four referees but I thought that it could be insightful to have some further thoughts on these issues as it may reveal some path forward in terms of conceptualizing and

modeling transcription and finding the molecular explanations of the observed context-dependent promoter dynamical properties. Would this make sense?

Response: We agree that this is a crucial point, and we have now extended the discussion section accordingly. Specifically, we now write:

“Here, we show that promoters can switch between distinct classes depending on context. We show that even under these relatively simple conditions, the same promoter can exhibit context-dependent scaling and induction behaviors. To describe this observation, we introduce the concept of context-dependent “manifestations”. The underlying number of molecular states of a promoter is potentially enormous – if we were to enumerate the combinatorial number of states based on nucleosome positions, TF occupancy at each binding site, binding of co-factors such as Mediator, SAGA, TFIID, RNA Polymerase II and numerous other factors, the number of discrete molecular states would be astronomically high. When we measure a dose-response, we likely observe only certain rate-limiting regimes or manifestations of the system. What we show here is that the particular observed rate-limiting manifestation is highly context-dependent and very distinct quantitative behaviors can be observed under different contexts – even in systems that are seemingly simple.

Does this mean that the concept of a few discrete promoter states is too strong an approximation to be useful? We suggest that this is not necessarily the case. Our analyses show that for a given context, a 3-state promoter architecture remains capable of quantitatively describing promoter dynamics. However, the specific 3 promoter states and their associated rates are in general dependent on Msn2 context. In other words, a complicated system can manifest itself in a simpler form under specific conditions. Comparing different manifestations across multiple input contexts can thus serve as an important means to unravel the overall complexity of promoter dynamics.”

3. {from second email: Indeed, it is important to address the points that the reviewers raised. Apologies, I see that my letter for some reason was jumping from #2 to #4. My paragraph #3 was in fact exactly to ask for that. I think most the comments are mostly addressable with amendments and clarification in the text.}

Response: Please see our responses to the reviewer comments from the other journal below.

4. The issue related to whether the system is ‘feedback-less’ is best addressed by some toning down and clarifying that formal exclusion of any feedback is difficult if not impossible.

Response: Yes, in retrospect we should have responded better to this point and we fully acknowledge this. What we meant was that Msn4 – a paralogue of Msn2 – is known to mediate feedback from Msn2 activation as shown by (AkhavanAghdam et al., 2016), and to our knowledge no other established forms of feedback exist. However, it is of course true that just because we have eliminated the only known-to-us feedback, does not mean that there are no other feedback systems that we just are not aware of. We hope the rephrasing in the sentence below more openly lays out this caveat:

“We note that the system is not subject to known feedback from Msn4 since Msn4 has been deleted in our system (AkhavanAghdam et al., 2016; Hansen and O’Shea, 2013; Hao and O’Shea, 2012), though we cannot rule out other forms of feedback.”

We have removed all claims of “feedback free” from the manuscript.

1st Editorial Decision

20th July 2020

Thank you again for submitting your work to Molecular Systems Biology. We are now satisfied with the changes made to the manuscript and I am pleased to inform you that we will be able to accept your paper for publication pending the minor amendments.

1st Authors' Response to Reviewers

15th Aug 2020

The Authors have made the requested editorial changes.

Accepted

25th Aug 2020

Dear Christoph,

Thank you again for sending us your revised manuscript. We are now satisfied with the modifications made and I am pleased to inform you that your paper has been accepted for publication.

Corresponding Author Name: Anders S. Hansen and Christoph Zechner

Manuscript Number: {has not been assigned yet}